# Coordinated active-reactive power optimization considering photovoltaic abandon based on second order cone programming in active distribution networks

Bo Peng[1], Yongjie Wang[1,2]*

**1** Faculty of Energy and Electrical Engineering, Qinghai University, Xining, China, **2** Qinghai Key Lab of Efficient Utilization of Clean Energy, Qinghai University, Xining, China

* wangyongjie@qhu.edu.cn

## Abstract

On the basis of predecessors' coordination optimization of active and reactive power in distribution network, For the necessity of the optimal operation in the distribution network, part of power generated from photovoltaic (PV) cannot be sold to users, and cannot enjoy subsidies. Similarly, the network loss in the power transmission will also bring a certain economic loss. This paper comprehensively considers the economic loss caused by the network loss and PV abandon of the distribution system, and establishes a model to minimize the economic loss. To solve this problem efficiently, the method of DistFlow equation and mixed integer second order cone programming (MISOCP) is used to solve the problem, in this method, the original mixed integer nonlinear programming non-convex problem is transformed into a convex problem, which makes the optimization problem easy to solve. The modified IEEE 33 and IEEE 69 distribution networks are tested by the above method. The optimized results are able to meet the target and have very small relaxation gaps, and the voltage level is also optimized. This coordinated optimization approach helps to optimize the economic operation for active distribution networks with PVs.

## 1 Introduction

With the depletion of fossil energy, the global demand for reducing carbon emission is on a constant increase. As a result, the distributed generation (DG) enjoys more and more popularity. Having the characteristics of local power absorption and being able to improve the stability of the power system [1], DG has become a sustainable mode for the development of renewable energy. However, the access of the DG to the distribution network has a significant impact on the voltage distribution of the power distribution network [2]. Considering that more and more DGs are connected to the distribution network, there are many challenges to the operation and management. There are various types of DGs accessing the distribution network. Photovoltaic (PV) is one of the most often used renewable sources [3]. Photovoltaic power generation has the role of promoting green energy transformation, protecting the ecological

**Funding:** This work was supported by the Open Fund of the State Key Laboratory f Power System Operation and Control(SKLD22KM10).

**Competing interests:** The authors have declared that no competing interests exist.

environment, and mitigating climate change, and it is an important way for China to realize the goal of carbon peaking and carbon neutrality [4]. In recent years, the installed capacity of PV generation in China is on a constant increase [5], which takes a great proportion and occupies an important position in the northwest power grid of China. Moreover, the installation rate of energy storage system, static reactive power compensation devices and capacitor banks in distribution network rise day by day [6, 7].

Active power optimization (APO) and reactive power optimization (RPO) of the distribution network are two important aspects of the optimization of the distribution network.

Zafar MH [8] use deep learning algorithms to optimize hybrid power generation scenarios with photovoltaics. Reference [9], used the optimized MPPT algorithm to optimize the power distribution network. Reference [10] use Runge Kutta Method algorithm achieves power optimization in the hot spot power generation (TEG) scenario.

Considering the R/X of the distribution system is large and P-Q is not decoupled [11], active power optimization of the distribution network will not only ameliorate the cost and other economic conditions, but also affect the voltage of the distribution network. In [12], PV inverters are used in the distribution system for reactive power optimization to reduce the loss of the system.

Based on the capability of the DG inverters to adjust the power factor, the power loss and network voltage level can be optimized through certain amount of reactive power provided by the inverter of DG [13–15]. Besides, many countries have adopted new grid regulations, in which the power factor for the new PV inverters should be 0.85 or higher [16–18]. Considering the characteristics of DG power factor, the accessed static reactive compensation (SVC) and capacitor banks (CBs), together with DG can be treated as a whole for coordinated active-reactive optimization.

In order to efficiently solve the power flow in the distribution network, the DistFlow equation is used for the radial distribution system. which is robust and efficient [19]. The power flow equation to solve the optimization of distribution network contains DGs, CBs, and SVCs. This problem is a mixed integer nonlinear problem, which is essentially non-convex, and difficult to solve. In order to effectively solve the aforementioned optimization problems, predecessors proposed a variety of algorithms for solving different objectives. Among the heuristic algorithms to solve this problem, there are genetic algorithm, particle swarm optimization (PSO) [20], sparrow search optimization algorithm(SSOA) [21],grey wolf optimizer(GWO) [22],meta-heuristic optimization algorithm(MOA) [23], etc. [24] adopted PSO—PRIM to optimize the low voltage distribution network design. PSO algorithm tries to improve the candidate solution by iterative method, but it is easy to fall into the local optima, and it also takes a long time to reach a desirable result. In [25], Krill Herd Algorithm (KHA) is adopted for the optimal configuration of DG to achieve the minimized network loss. In [26, 27], the meta-heuristic algorithm is adopted to minimize the investment cost of distribution network. In [28], the multi-objective method is adopted to minimize the actual network loss and net re-active power flow through the optimal configuration and classification of DGs and capacitors. These kinds of heuristic algorithm are relatively easy to implement, and therefore, it has attracted wide attention. When solving nonlinear non-convex problems of mixed integers, the analytic algorithm is faster than the heuristic algorithm, and has a rigorous demonstration process. However, as for the non-convex power flow equation of distribution network, traditional methods such as interior point method and gradient descent can only guarantee to get the local optimal solution [11]. In article [29], semi-defined programming is used to relax the original model into a convex model. In [30], the whale optimization algorithm is used to optimize the power loss and voltage characteristics of the non-convex scenario. Compared with the above methods, the

SOCP method, which relaxes the flow equation into a convex problem, has a simpler calculation process, and reduces the number of dummy variables and simplifies the complexity. Based on the above discussion, the SOCP method [31, 32] for the branch flow model is more convenient and is of higher quality. [33, 34] analyzed the feasibility of relaxation properties after second order cone (SOC) relaxation being carried out. When the gap between the obtained results and the original feasible region is small, the results have a high accuracy. [35] used the MISOCP to solve the optimal configuration of DG. In [36], the SOCP is used to provide a day-ahead optimal scheduling scheme considering two-way power flow for gas and electric power systems. In [37], the SOCP is used to optimize the distribution of energy storage capacity in the distribution network. In [38], two objectives of minimum network loss and generation cost of power system are established by using SOCP. These papers all use the SOCP method to relax the non-convex problems into convex problems, and have considerable accuracy.

The traditional optimization of parameter configurations usually takes the minimum network loss as the optimization goal. Nowadays, as more new devices are added and more versatile energy sectors like heating networks are integrated in the electric power network [39–42], the optimization goal of the distribution network becomes diverse. In order to improve the voltage distribution, [43] establishes an active and reactive power optimization model to minimize the operation cost. Considering that PV power generation is the main form of medium and low voltage DG, PV is selected as a typical DG in this paper. In [44], the total operating cost is taken as the objective function, and the optimization model of the distribution network with wind and solar clusters is constructed by considering various constraints such as network loss cost, unit operating cost, and energy storage equipment constraints, demand response constraints, etc. In [45], the optimization of PV to distribution network is discussed, and the optimization characteristics such as objective function, variables and constraints are summarized.

Through the analysis of various studies, it is found that for active photovoltaic distribution networks, few studies have comprehensively analyzed the economic loss of distribution networks considering the abandonment of photovoltaic power. The main economic loss of distribution networks containing photovoltaic power is the loss of photovoltaic abandonment and the economic loss of line loss in this distribution network. Therefore, we plan to focus the optimization objective on the economic loss caused by photovoltaic loss and photovoltaic abandonment. However, this optimization scenario is non-convex and difficult to solve using conventional planning algorithms or heuristic algorithms. Therefore, in this active-passive coordinated optimization economic loss scenario, we introduce MISOCP to solve this problem, which can convert non-convex problems into convex problems, which are easier to solve and easier to obtain the global optimal solution. In order to verify its accuracy, we finally analyzed the degree of relaxation, and the results were accurate enough.

For the optimal operation of distribution network, a part of the output of DG is abandoned. This part of the abandoned output, together with the transmission loss, will bring certain economic loss for the whole system. The price of PV grid usually falls along with the cost of PV products, and some countries have policies to provide a certain amount of subsidy [46]. Considering the current development of PV industry in Qinghai Province of China is relatively mature, this paper developed a general economic model for network loss and PV abandon. Therefore, a modified 33-node distribution system with PV DG is designed for the optimization containing the network loss and PV abandon according to certain weight. The DistFlow equation-based MISOCP is used for the comprehensive optimization of the static configuration of the distribution network with DGs, by minimizing the sum of PV abandon and

network loss with respect to certain weights in a unit time. Moreover, based on the above method, another group of 69-node distribution system with PV generations was modified to carry out the same analysis and verification. And the simulation of the calculation example satisfies the demand, which has reference significance for the economic operation of distribution network.

## 2 Methods and constraints of model establishment

### 2.1. DistFlow branch flow model

For illustration purposes, a sample distribution network with a radial shape is introduced as shown in the Fig 1:

The Distflow form of the power flow equation is:

For any node $j$:

$$\sum_{i\in u(j)}\left(P_{ij} - \frac{(P_{ij})^2 + (Q_{ij})^2}{(V_i)^2}r_{ij}\right) + P_j = \sum_{k\in v(j)}P_{jk}$$

$$\sum_{i\in u(j)}\left(Q_{ij} - \frac{(P_{ij})^2 + (Q_{ij})^2}{(V_i)^2}x_{ij}\right) + Q_j = \sum_{k\in v(j)}Q_{jk}$$

(1)

For the branch $ij$:

$$(V_j)^2 = (V_i)^2 - 2\left(r_{ij}P_{ij} + x_{ij}Q_{ij}\right) + \left((r_{ij})^2 + (x_{ij})^2\right)\frac{(P_{ij})^2 + (Q_{ij})^2}{(V_i)^2}$$

(2)

In the above formula, set $u(j)$ represents the set of the head nodes of the branch with $j$ as the end node in the power grid. Similarly, set $v(j)$ represents the set of end nodes of the branch with $j$ as the head node in the power grid. $P_{ij}$ and $Q_{ij}$ respectively represent the active and reactive power at the first end of branch $ij$. $V_i$ represents the voltage amplitude of node $i$. $r_{ij}$ and $x_{ij}$ represent the resistance and reactance of branch $ij$.

For the node j:

$$P_j = P_{j,\mathrm{DG}} - P_{j,\mathrm{d}}$$
$$Q_j = Q_{j,\mathrm{DG}} + Q_{j,\mathrm{com}} - Q_{j,\mathrm{d}}$$

(3)

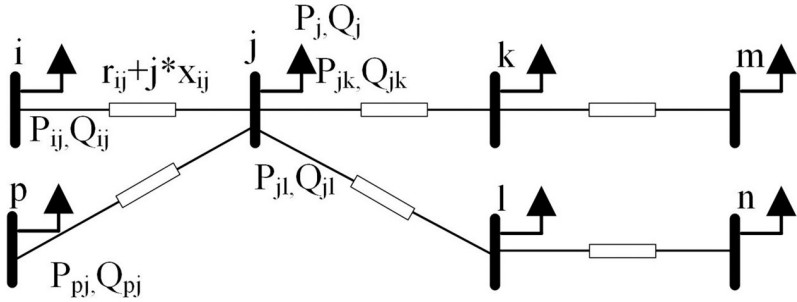

**Fig 1. A 7-node radial system.**

In the above formula, $P_j$ and $Q_j$ are the net injection of active and reactive power of node $j$. $P_{j,DG}$ and $Q_{j,DG}$ are respectively the active and reactive power of DG connected to node $j$, $P_{j,d}$ and $Q_{j,d}$ are the active and reactive power of the load connected to node $j$ and $Q_{j,com}$ is the reactive power of the reactive compensation device connected to node $j$, such as CBs, SVCs, etc.

## 2.2. Objective function based on Distflow

The objective function is defined as:

$$\min C_1 \sum_{i=1}^{N_{bus}} \sum_{j \in v(i)} r_{ij} |I_{ij}|^2 + C_2 \sum_{k=1}^{N_{DG}} (P_{DGk\,\max} - P_{DGk}) \tag{4}$$

In this formula, $N_{bus}$ and $N_{DG}$ are the number of nodes and the number of DGs respectively, $P_{DGk\max}$ is the maximum active power that can be generated by each DG, and $P_{DGk}$ is the actual active power of grid-connection for the current period of time. The definition of set $v(i)$ is similar to Eq (1), representing the set of end nodes of the branch starting with $i$ in the power grid. $|I_{ij}|^2$ is the square of the amplitude of the branch current, which can be obtained from Eq (2):

$$|I_{ij}|^2 = \frac{(P_{ij})^2 + (Q_{ij})^2}{(V_i)^2} \tag{5}$$

The objective function consists of two parts: transmission line loss and photovoltaic module abandonment loss. These two parts are selected as representatives of economic losses based on their significant impact on the economy and efficiency of the power system. In the objective function, the first part with the coefficient $C_1$ represents the economic loss caused by the power transmission loss of all branches of the whole network. Transmission line losses are mainly due to energy losses caused by resistance and other factors during the transmission process. Specifically, when the current flows in the transmission line, it encounters resistance, causing part of the electrical energy to be converted into heat energy, resulting in losses. This loss is proportional to the square of the current and is related to the resistance of the line. Reducing transmission line losses can not only improve the transmission efficiency of the power grid, but also reduce the operating costs of power generation and transmission, thereby achieving improved economic benefits. The second part with the coefficient $C_2$ represents the economic loss from the subsidies and the direct loss due to unsold electricity from PVs. The photovoltaic module abandonment loss refers to the amount of power that has to be abandoned because the photovoltaic power generation capacity exceeds the actual load demand or the grid's absorption capacity is insufficient. The reason behind this loss is the volatility and intermittent characteristics of photovoltaic power generation, which makes the photovoltaic power generation exceed the demand or carrying capacity of the grid in certain periods. The abandonment phenomenon leads to the waste of potential power generation, which in turn affects the economy and sustainability of photovoltaic power generation. By optimizing the operation of the power grid and reducing the abandonment loss of photovoltaic modules, renewable energy can be used more effectively and the overall economic benefits can be improved.

The selection of transmission line losses and photovoltaic module abandonment losses as representatives of economic losses aims to fully reflect the main economic factors in the operation of the distribution network. Transmission line losses involve the problem of energy transmission efficiency in traditional power systems, while photovoltaic module

abandonment losses reflect the problem of renewable energy utilization in modern power systems. By optimizing these two parts, the overall economic benefits of the distribution network can be maximized, while promoting the sustainable development of the power system.

In conclusion, $C_1$ is the coefficient of network loss to indicate a particular period of time the average electricity price in one hour, $C_2$ is the coefficient of PV abandon to indicate a particular period of time the average electricity price in one hour plus the subsidies per kilowatt. That is

$C_1$ = electricity price per kilowatt in a given hour.

$C_2$ = electricity price per kilowatt in a given hour + subsidy per kilowatt.

The sum of two parts with coefficients represents the loss to the overall economic operation of the system due to PV abandon and network loss in a given hour. In China, the price of PV electricity is jointly subsidized by the national government and local government. The price of electricity generated by the distributed PV power station can be different in different areas, but the national subsidy for each generation unit is the same. In 2023, the national average electricity price in China will be about 0.594¥/kWh. In line with China's West-to-East Power Transmission Policy, Qinghai Province integrates various types of power generation locally, resulting in high power generation and relatively low electricity charges. On this basis, together with the "guided price for electricity" policy carried out by Qinghai province in 2022, the actual price range for bidding in 2023 is 0.2427–0.4492¥/kWh. By 2023, the average electricity price in Qinghai Province, China is 0.390¥/kWh, so we set it as the network loss coefficient for this study. Qinghai Province, China is one of the main locations of China's photovoltaic power generation industry, so its subsidies are relatively high. In 2023, the official price of photovoltaic grid-connected power in Qinghai Province is 0.301¥/kWh, and the combined grid-connected power price subsidy of the local government and the national government is 0.050¥/kWh. The electricity price in an hour according to the local electricity price in Qinghai Province, the PV feed-in tariff and the feed-in tariff mitigation are shown in Table 1.

The price of electricity from the power grid includes industrial and commercial price, large industrial price and residential price, and varies greatly for different users on the user side. The network loss means the economic loss of electricity that the grid fails to sell to users, and the PV abandon means the economic loss from the abandoned power in PV. The feed-in tariff mitigation is for the feed-in PV. The coefficient of the two factors are set as 0.390 and 0.351 respectively. That is:

$$\begin{aligned} C_1 &= 0.390 \\ C_2 &= 0.301 + 0.050 = 0.351 \end{aligned} \tag{6}$$

Then we normalized $C_1$ and $C_2$, and were treated 0.526 and 0.474. Therefore, the sum of the first and the second part multiplied by the coefficient of 0.526 and 0.474 represents the general model of economic operation.

**Table 1. The price information for the PVs and the grid.**

| Electricity Price (¥/kWh) | PV feed-in tariff(¥/kWh) | Feed-in tariff mitigation(¥/kWh) |
|---|---|---|
| 0.390 | 0.301 | 0.050 |

## 2.3. Constraints

1. Power balance constraints:(1), (2), (3). In distribution network optimization models, power balance constraints are imposed to ensure that the power supply at each node in the grid matches the demand. These constraints ensure that at any given moment, the total amount of power generation in the grid equals the total amount of load demand, plus losses in the system.

2. Safety constraints of node voltage and branch current:

In the distribution network optimization model, node voltage and branch current constraints are imposed to ensure the safety, stability and reliability of the power grid. These constraints ensure that the voltage is within the specified range, prevent equipment damage and power quality degradation, and ensure that the current does not exceed the rated capacity of the equipment and lines, prevent overload and overheating, thereby ensuring the safe operation of the power grid and the long life of the equipment. Through these constraints, the optimization model ensures that all parameters of the power grid are always within the safe range while pursuing economic benefits. The constraints are as follows:

$$
\begin{aligned}
V_i^{\min} \leq V_i \leq V_i^{\max} \\
I_{ij} \leq I_{ij}^{\max}
\end{aligned}
\tag{7}
$$

In the formula above, $V_i$ is the voltage amplitude of node $i$, $V_i^{\min}$ and $V_i^{\max}$ respectively are the maximum and minimum limits of voltage amplitude, $I_{ij}$ is the current amplitude of the branch, and $I_{ij}^{\max}$ is the overload critical current amplitude of the branch. In this simulation study, the upper and lower limits of node voltage are set to 1.03 and 0.97 times. The upper and lower limits of branch current are 1.2kA and -1.2kA.

3. Power constraint

In the distribution network optimization model, active and reactive power balance constraints are key to ensure that the power supply and demand of each node in the power grid are matched. These constraints ensure the stable operation and power quality of the power grid. In order to suppress the influence of the power fluctuation of the active distribution network on the transmission network, the power constraints of the root node of the distribution network need to be taken into account, that is:

$$
\begin{aligned}
P_0^{\min} \leq P_0 \leq P_0^{\max} \\
Q_0^{\min} \leq Q_0 \leq Q_0^{\max}
\end{aligned}
\tag{8}
$$

In this formula, $P_0$ is the power into the distribution network from the root node. $P_0^{\min}$ and $P_0^{\max}$ are the upper and lower bounds of active power set by the control center respectively. The constraints of reactive power can be calculated similarly.

4. Constraint of discrete reactive power compensation device

In the distribution network optimization model, the constraints of discrete reactive compensation devices are to ensure that the reactive compensation devices are within a reasonable operating range and to correctly enable or disable these devices during the optimization process to achieve reactive power balance and voltage stability. The switching state of CB is a

discrete decision variable, and the following linearized model is adopted in this paper:

$$\begin{aligned} t_i Q_{i,\text{com}}^{\text{step}} &= Q_{i,\text{com}}^{dis} \\ 0 \leq t_i &\leq n \\ t_i &\in N^* \end{aligned} \tag{9}$$

In the formula, $t_i$ is an integer variable, $Q_{i,\text{com}}^{\text{step}}$ is the step size of each CB, $Q_{i,\text{com}}^{dis}$ is the output of the $i$th CB, and $n$ is the maximal step number of each CB. This simulation has four gears: 0, 1, 2, 3.

5. Constraint of continuous reactive power compensation device

In the distribution network optimization model, the constraints of continuous reactive power compensation devices (such as static VAR compensators SVC or static VAR generators STATCOM) are to ensure that these devices provide appropriate reactive power within their adjustable range to achieve reactive power balance and voltage stability. The constraints are as follows:

$$Q_{i,\text{com}}^{\text{min}} \leq Q_{i,\text{com}}^{con} \leq Q_{i,\text{com}}^{\text{max}} \tag{10}$$

In this formula, $Q_{i,\text{com}}^{con}$ represents the output of a certain continuous reactive compensation device. $Q_{i,\text{com}}^{\text{min}}$ and $Q_{i,\text{com}}^{\text{max}}$ are the upper and lower limits of outputs for SVCs.

The above equation reflects the reactive power compensation capacity constraints of compensating devices with continuously and independently adjustable power (SVCs, etc.).

6. DG operation constraint

In the distribution network optimization model, the operating constraints of DG are to ensure that these power sources operate within their technical and physical limitations and coordinate their operation with the grid to ensure the stability and reliability of the power system. The constraints are as follows:

$$\begin{aligned} 0 \leq P_{i,\text{DG}} &\leq P_{i,\text{DG}}^{pre} \\ 0 \leq Q_{i,\text{DG}} &\leq Q_{i,\text{DG}}^{\text{pre}} \\ Q_{i,\text{DG}}^{\text{pre}} &= P_{i,\text{DG}}^{\text{pre}} \tan \varphi \end{aligned} \tag{11}$$

The remaining variables in formula (11) are already defined above. $P_{i,\text{DG}}$ and $Q_{i,\text{DG}}$ represent the active power output and reactive power output of a certain DG respectively. The steady state operation of DG in this paper adopts PQ type. DG accesses the power grid through power electronic equipment or conventional rotary motor interface, and the power of DG has achieved the independent adjustment of both active and reactive power. To make full use of distributed clean energy, this paper uses the MPPT as the DG operation mode. The active and reactive power output by DG are set as an adjustable variable within the interval $[0, P_{i,\text{DG}}^{pre}]$ and $[0, Q_{i,\text{DG}}^{pre}]$ separately where $Q_{i,\text{DG}}^{\text{pre}} = P_{i,\text{DG}}^{\text{pre}} \tan \varphi$, and $\varphi$ is the power factor angle, considering the flexibility of the power electronic equipment of PV inverter interface.

To sum up, the control variables in the reactive voltage optimization problem of the active distribution networks are the distributed generation $Q_{i,\text{DG}}$ and the operational power of continuous and discrete reactive power compensation device $Q_{i,\text{com}}^{con}, Q_{i,\text{com}}^{\text{dis}}$. The above problem is a typical mixed integer nonlinear non-convex programming problem, which belongs to NP-hard problem.

## 3 Solution method

SOCP is a powerful optimization technique for minimizing linear objective functions under complex constraints. SOCP constraints include second-order cone constraints and linear equality constraints, which can describe complex situations such as norm restrictions. Compared with linear programming and quadratic programming, SOCP can handle a wider range of problem types.

A typical application scenario involves solving non-convex problems. Non-convex problems are usually difficult to solve directly to the global optimal solution due to their complexity. In this case, mathematical derivation and relaxation procedures are particularly important. By converting the non-convex constraints in the original problem into second-order cone constraints, we relax the problem into a convex optimization problem. This relaxation process makes the originally difficult non-convex problem solvable. Specifically, through this conversion, the non-convexity of the original problem is simplified, allowing us to efficiently find the global optimal solution of the problem using the existing SOCP solution algorithm.

This non-convex-to-convex relaxation process not only improves the solution efficiency, but also ensures the accuracy of the solution, making SOCP a powerful tool for solving complex optimization problems. In practical applications, this method is widely used in engineering, finance, and machine learning, and has achieved significant application value by accurately modeling and effectively solving optimization problems under complex constraints. Next, we analyze SOCP step by step and apply it to the relaxation work in the current scenario.

### 3.1. The standard form of SOCP

The standard form of SOCP is defined as:

$$\min_{x_i}\{\mathbf{c}^T\mathbf{x}|\mathbf{A}\mathbf{x}=\mathbf{b}, x_i \in K, i=1,2\ldots,N\} \tag{12}$$

In the above formula, $\mathbf{x} \in \mathbf{R}_N$ is a variable, $\mathbf{b} \in \mathbf{R}_M$, $\mathbf{c} \in \mathbf{R}_N$, $\mathbf{A}_{M \times N} \in \mathbf{R}_{M \times N}$ are coefficient constants, and $K$ is a second order cone or a rotating second order cone of the following form:

a) second order

$$K = \{x_i \in \mathbf{R}_N | y \geq \sqrt{\sum_{i=1}^{N} x_i^2}, y \geq 0\} \tag{13}$$

b) rotating second order

$$K = \{x_i \in \mathbf{R}_N | yz \geq \sqrt{\sum_{i=1}^{N} x_i^2}, yz \geq 0\} \tag{14}$$

SOCP can be regarded as an extension of linear programming, which is essentially a convex optimization. SOCP has positive properties such as the optimality of solutions and computational efficiency. The existing CPLEX, GROBI, MOSEK and other algorithm packages can be used to achieve favorable results, and many SOCP-based optimization problems can be completed within a polynomial time.

### 3.2. MISOCP for optimization of distribution network

Considering the strong non-convex form of formulas (1) and (2) the above problem belongs to NP-hard problem, where finding the optimal solution is difficult and the solving efficiency

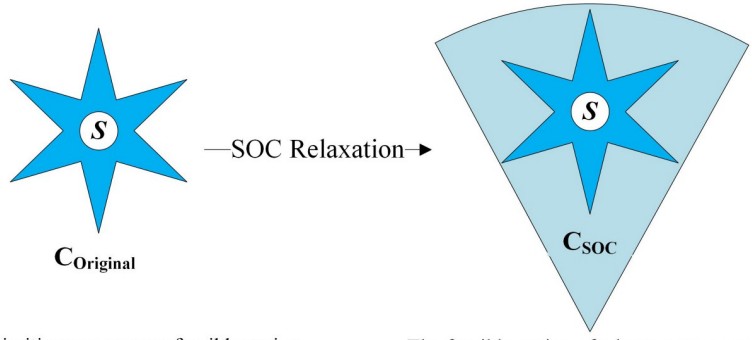

**Fig 2. Simplified schematic diagram of feasibility domain.** Primitive non-convex feasible region. The feasible region of a loose convex cone.

is low. However, the problem can be relaxed using the SOCP. Two new variables can be introduced: squared voltage amplitude $v_{2,i}$ and squared branch current amplitude $i_{2,ij}$:

$$v_{2,i} = (V_i)^2$$
$$i_{2,ij} = |I_{ij}|^2 = \frac{(P_{ij})^2 + (Q_{ij})^2}{(V_i)^2} \tag{15}$$

Replace the related terms in both the objective function and constraints with the above variables, and the objective function becomes $\min C_1 \sum_{i=1}^{N_{bus}} \sum_{j \in v(i)} r_{ij} i_{2,ij} + C_2 \sum_{k=1}^{N_{DG}} (P_{DGk\,\max} - P_{DGk})$. Then we can add $i_{2,ij} = \frac{(P_{ij})^2 + (Q_{ij})^2}{v_{2,i}}$ to the constraint. Under a set of sufficient conditions, such as the strict increment function of the objective function being $i_{2,ij}$, the above equation can be

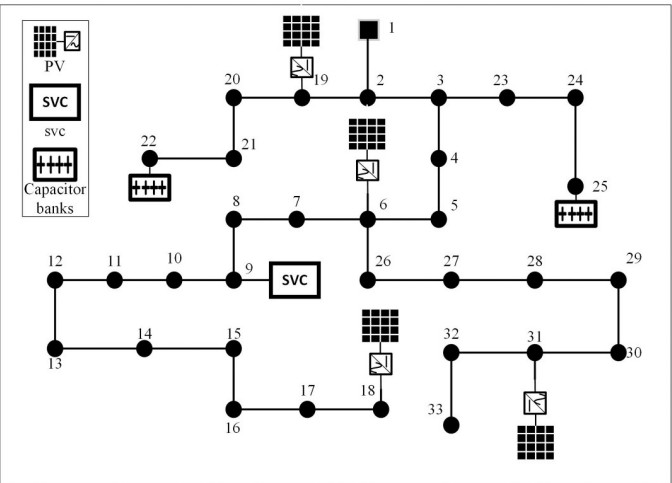

**Fig 3. Modified IEEE 33 distribution network.**

**Table 2. The load of each bus and the impedance of each branch in modified IEEE 33.**

| Bus number | $P_d$ (kW) | $Q_d$(kVar) | branch from-to | Impedance($\Omega$) |
|---|---|---|---|---|
| 1 | 0 | 0 | 1–2 | 0.0922+j0.047 |
| 2 | 100 | 60 | 2–3 | 0.493+j0.2511 |
| 3 | 90 | 40 | 3–4 | 0.366+j0.1864 |
| 4 | 120 | 80 | 4–5 | 0.3811+j0.1941 |
| 5 | 60 | 30 | 5–6 | 0.819+j0.707 |
| 6 | 60 | 20 | 6–7 | 0.1872+j0.6188 |
| 7 | 200 | 100 | 7–8 | 0.7114+j0.2351 |
| 8 | 200 | 100 | 8–9 | 1.03+j0.74 |
| 9 | 60 | 20 | 9–10 | 1.044+j0.74 |
| 10 | 60 | 20 | 10–11 | 0.1966+j0.065 |
| 11 | 45 | 30 | 11–12 | 0.3744+j0.1238 |
| 12 | 60 | 35 | 12–13 | 1.468+j1.155 |
| 13 | 60 | 35 | 13–14 | 0.5416+j0.7129 |
| 14 | 120 | 80 | 14–15 | 0.591+j0.526 |
| 15 | 60 | 10 | 15–16 | 0.7463+j0.545 |
| 16 | 60 | 20 | 16–17 | 1.289+j1.721 |
| 17 | 60 | 20 | 17–18 | 0.732+j0.574 |
| 18 | 90 | 40 | 2–19 | 0.164+j0.1565 |
| 19 | 90 | 40 | 19–20 | 1.5042+j1.3554 |
| 20 | 90 | 40 | 20–21 | 0.4095+j0.4784 |
| 21 | 90 | 40 | 21–22 | 0.7089+j0.9373 |
| 22 | 90 | 40 | 3–23 | 0.4512+j0.3083 |
| 23 | 90 | 50 | 23–24 | 0.898+j0.7091 |
| 24 | 420 | 200 | 24–25 | 0.896+j0.7011 |
| 25 | 420 | 200 | 6–26 | 0.203+j0.1034 |
| 26 | 60 | 25 | 26–27 | 0.2842+j0.1447 |
| 27 | 60 | 25 | 27–28 | 1.059+j0.9337 |
| 28 | 60 | 20 | 28–29 | 0.8042+j0.7006 |
| 29 | 120 | 70 | 29–30 | 0.5075+j0.2585 |
| 30 | 200 | 600 | 30–31 | 0.9744+j0.963 |
| 31 | 150 | 70 | 31–32 | 0.3105+j0.3619 |
| 32 | 210 | 100 | 32–33 | 0.341+j0.5302 |
| 33 | 60 | 40 | | |

transformed as follows:

$$i_{2,ij} \geq \frac{\left(P_{ij}\right)^2 + \left(Q_{ij}\right)^2}{v_{2,i}} \tag{16}$$

**Table 3. Optimal solution in modified IEEE 33.**

| Solution time (s) | Objective Function: $C_1$*Network loss+ $C_2$*PV abandon (¥) | | |
|---|---|---|---|
| | 0.526*Network loss | 0.474 *PV abandon | Sum |
| 1.07 | 15.16 | 879.83 | 894.99 |

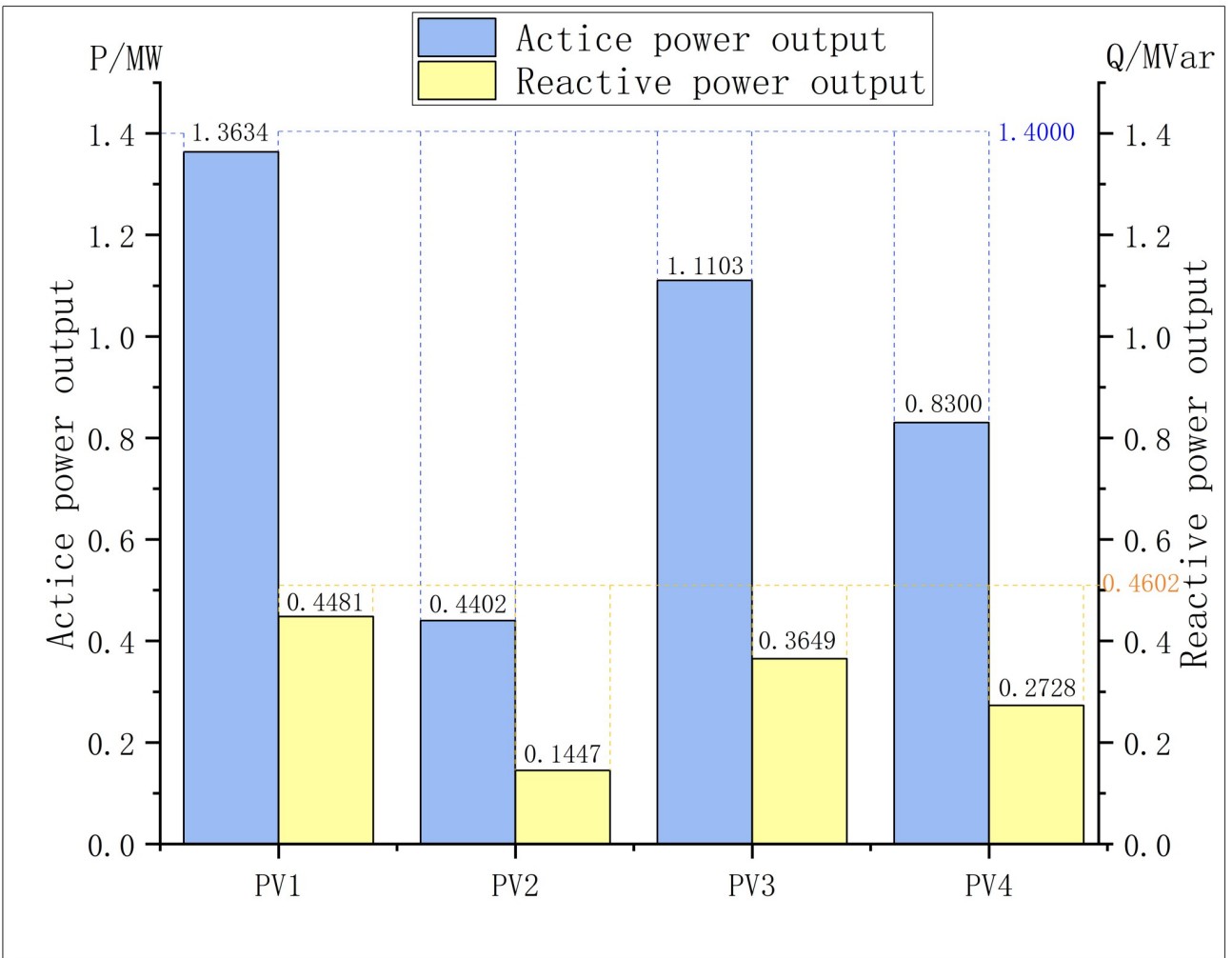

**Fig 4. PV outputs in modified IEEE 33.**

Then, after equivalent transformation, Eq (15) is written in the form of a standard SOC, that is:

$$\left\| \begin{array}{c} 2P_{ij} \\ 2Q_{ij} \\ i_{2,ij} - v_{2,i} \end{array} \right\|_2 \leq i_{2,ij} + v_{2,i} \tag{17}$$

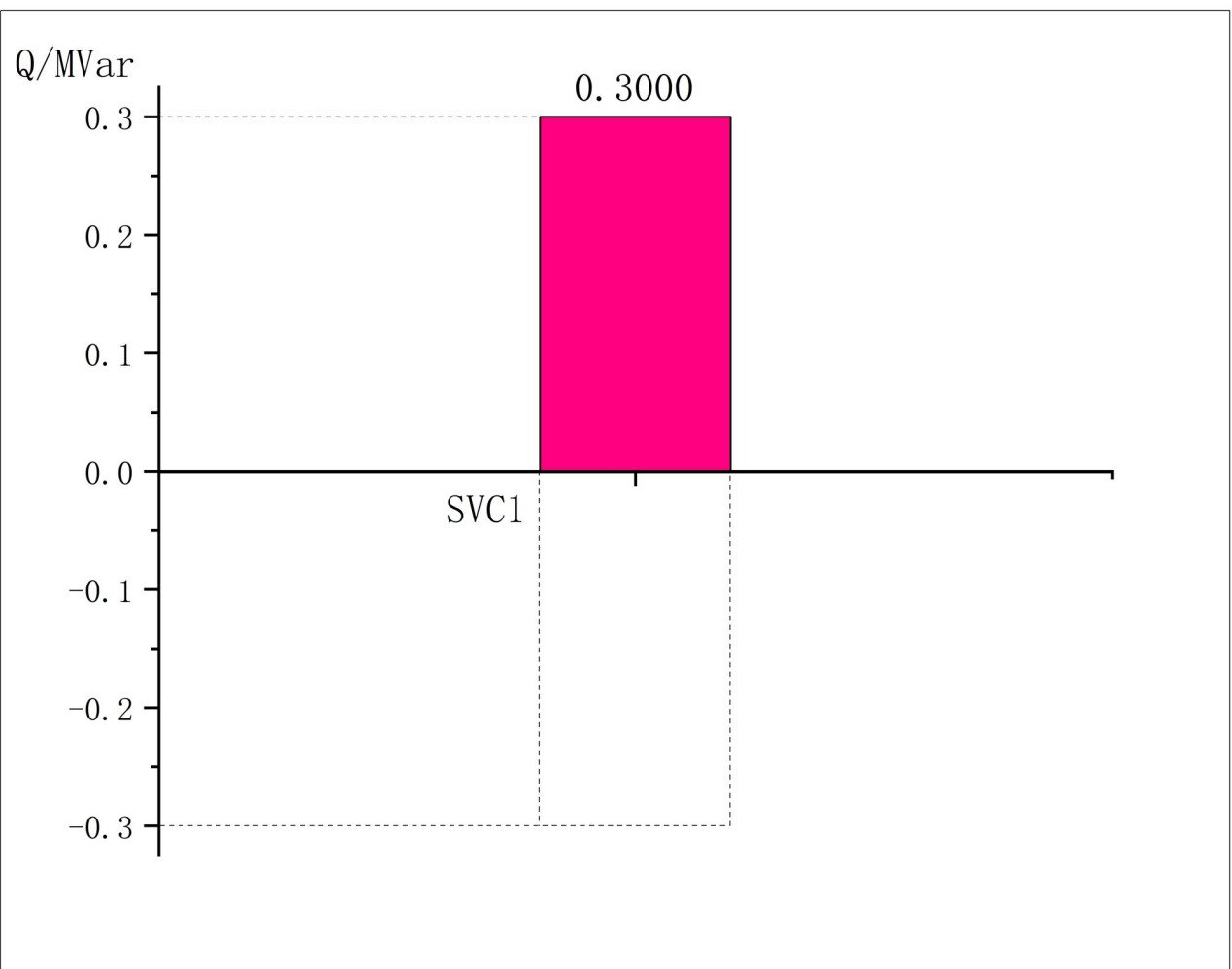

**Fig 5. SVC output in modified IEEE33.**

Thus, the power flow equation can be transformed into the following form:

$$\sum_{i \in u(j)} (P_{ij} - i_{2,ij} r_{ij}) + P_j = \sum_{k \in v(j)} P_{jk}$$

$$\sum_{i \in u(j)} (Q_{ij} - i_{2,ij} x_{ij}) + Q_j = \sum_{k \in v(j)} Q_{jk}$$

$$v_{2,j} = v_{2,i} - 2(r_{ij} P_{ij} + x_{ij} Q_{ij}) + \left( \left( r_{ij} \right)^2 + \left( x_{ij} \right)^2 \right) i_{2,ij} \qquad (18)$$

$$\left\| \begin{array}{c} 2P_{ij} \\ 2Q_{ij} \\ i_{2,ij} - v_{2,i} \end{array} \right\|_2 \leq i_{2,ij} + v_{2,i}$$

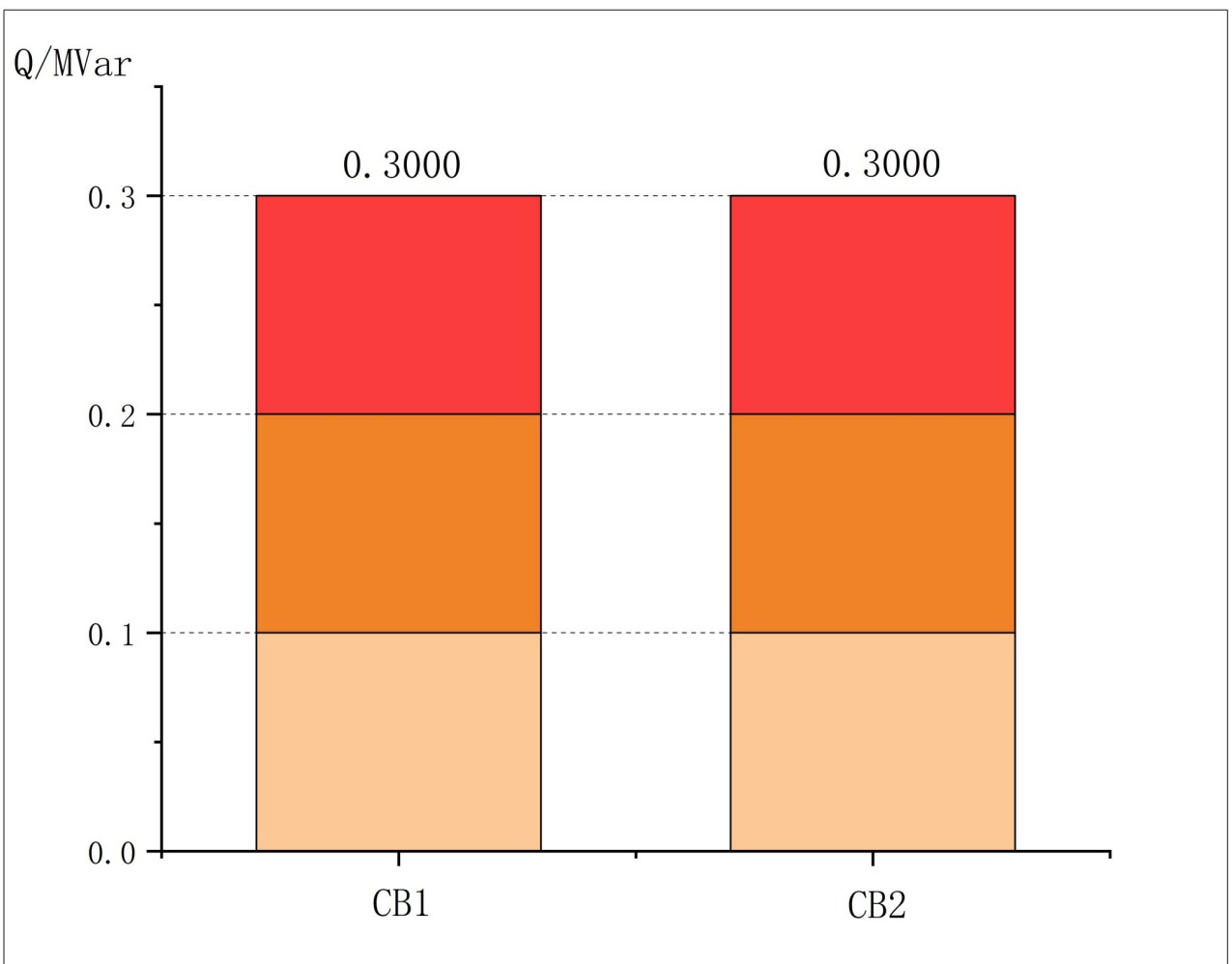

**Fig 6. CBs output in modified IEEE33.**

Then the original reactive power optimization problem is finally transformed into the following model:

$$\min w_1 \sum_{i=1}^{N_{bus}} \sum_{j \in v(i)} r_{ij} i_{2,ij} + w_2 \sum_{k=1}^{N_{DG}} (P_{DGk\,\max} - P_{DGk})$$

$$s.t.(3),(6),(7),(8),(9),(10),(11),(18)$$

$$(19)$$

### 3.3. Solvability analysis

For the following mathematical problems:

$$\min \quad f(x)$$
$$s.t. \quad g_i(x) \le 0, i = 1, 2, \ldots, m$$
$$\qquad h_j(x) = 0, j = 1, 2, \ldots, l$$

$$(20)$$

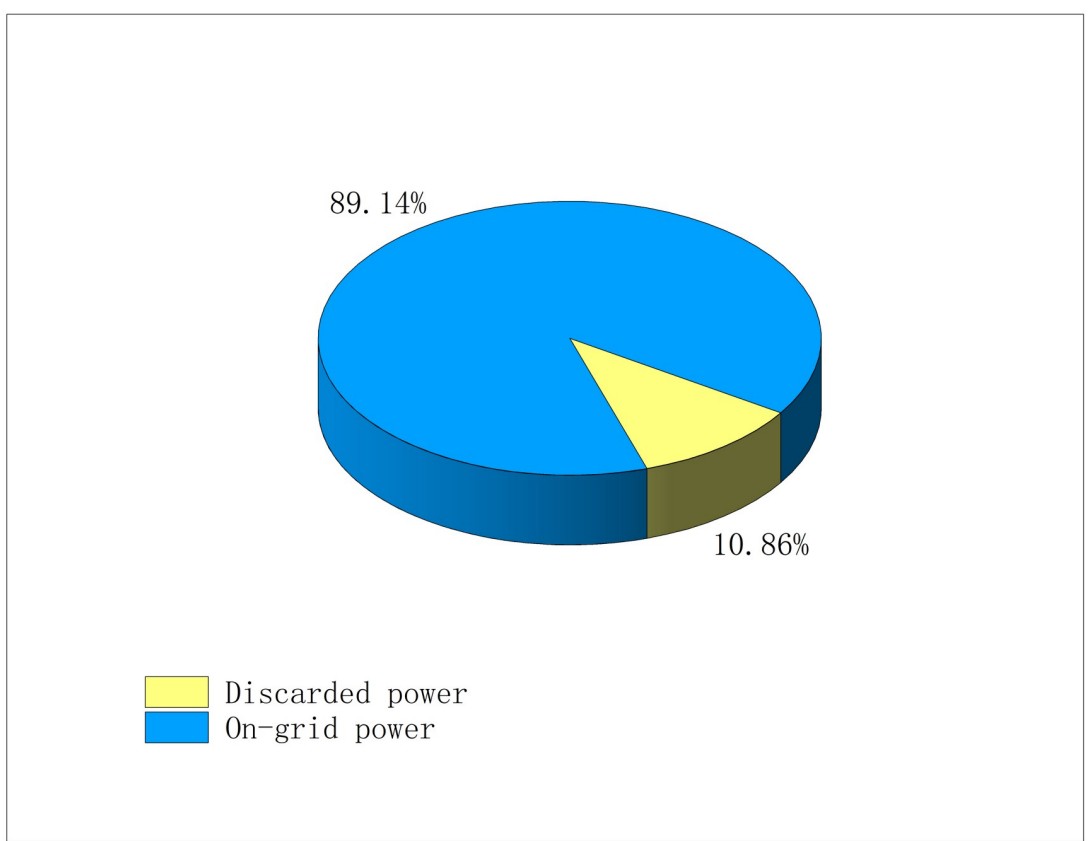

**Fig 7. The proportions of abandoned and on-grid power from all PVs.**

The criteria for this model to be convex programming are: the objective function *f(x)* being a convex function (the Hessen matrix is semi-positive definite), all equations in the constraint $h_j(x)$ being linear functions, and all Hessen matrix of inequality constraint $g_i(x)$ being semi-positive definite.

For the optimization model proposed in this paper, if discrete control variables such as energy storage and reactive power compensation device are not taken into account, the objective function, all equality constraints, and all inequality constraints except Eq (16) can be all considered as linear. The mathematical form of Eq (16) can be abstractly written as $\sqrt{x_1^2 + x_2^2 + x_3^2} \leq x_4$. This form of inequality satisfies the definition of the SOC, whose Hessen matrix (the second derivative matrix) is:

$$\begin{bmatrix} 1/(x_1^2 + x_2^2 + x_3^2)^{(1/2)} - x_1^2/(x_1^2 + x_2^2 + x_3^2)^{(3/2)} & -(x_1 * x_2)/(x_1^2 + x_2^2 + x_3^2)^{(3/2)} & -(x_1 * x_3)/(x_1^2 + x_2^2 + x_3^2)^{(3/2)} & 0 \\ -(x_1 * x_2)/(x_1^2 + x_2^2 + x_3^2)^{(3/2)} & 1/(x_1^2 + x_2^2 + x_3^2)^{(1/2)} - x_2^2/(x_1^2 + x_2^2 + x_3^2)^{(3/2)} & -(x_2 * x_3)/(x_1^2 + x_2^2 + x_3^2)^{(3/2)} & 0 \\ -(x_1 * x_3)/(x_1^2 + x_2^2 + x_3^2)^{(3/2)} & -(x_2 * x_3)/(x_1^2 + x_2^2 + x_3^2)^{(3/2)} & 1/(x_1^2 + x_2^2 + x_3^2)^{(1/2)} - x_3^2/(x_1^2 + x_2^2 + x_3^2)^{(3/2)} & 0 \\ 0 & 0 & 0 & 0 \end{bmatrix} \quad (21)$$

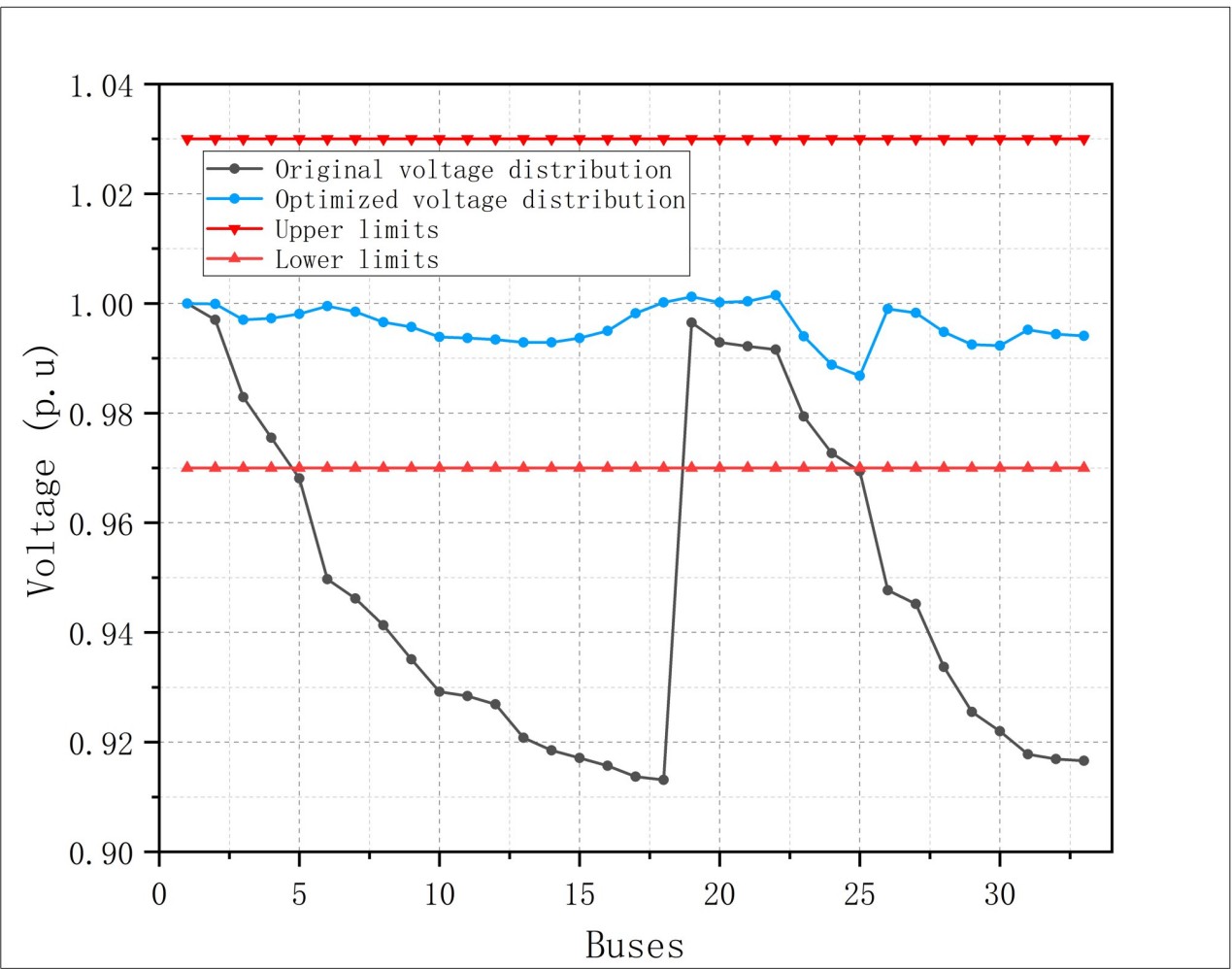

**Fig 8. Bus voltage amplitude before and after optimization in modified IEEE 33.**

The eigenvalues of the above Hessen matrix are:

$$\begin{bmatrix} 0 \\ 0 \\ \left(x_1^4 + 2*x_1^2*x_2^2 + 2*x_1^2*x_3^2 + x_2^4 + 2*x_2^2*x_3^2 + x_3^4\right)/\left(x_1^2 + x_2^2 + x_3^2\right)^{(5/2)} \end{bmatrix} \quad (22)$$

It is not difficult to find that the eigenvalues of the matrix are all non-negative, that is, the Hessen matrix is semi-positive definite, which indicates that the feasible region of the SOC is convex. In other words, the above problem is a SOC convex programming problem.

After adding discrete variables, this model is no longer a strictly convex programming problem, but the convexification of the power flow equation, a strong non-convex source, changes the mathematical properties of the problem, and the solvability and optimality are improved accordingly.

If integer variables are excluded, as shown in the Fig 2, the feasible domain of the original problem $C_{original}$ will be relaxed into a feasible domain $C_{soc}$ that is a convex SOC. At this point, the original problem essentially becomes a convex programming. Due to the introduction of

**Fig 9. Optimal solution and relaxation gap changes under different weighting factors.**

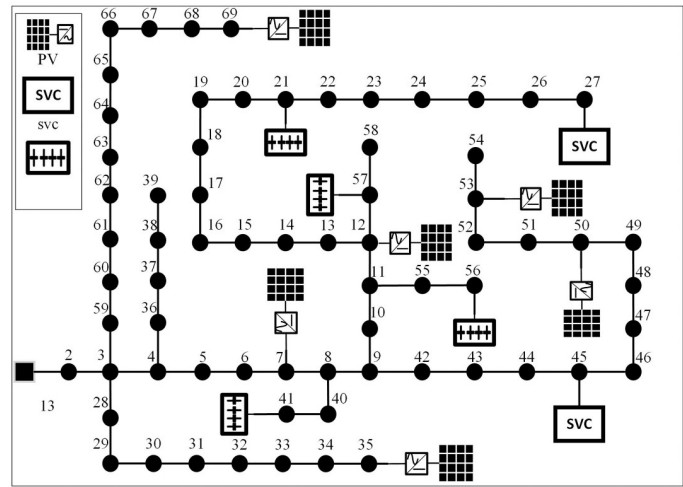

**Fig 10. Modified IEEE 69 distribution network.**

**Table 4. The load of each bus and the impedance of each branch in modified IEEE 69.**

| Bus number | $P_d$ (kW) | $Q_d$(kVar) | branch from-to | Impedance($\Omega$) |
|---|---|---|---|---|
| 1 | 0 | 0 | 1–2 | 0.0050+j0.0012 |
| 2 | 0 | 0 | 2–3 | 0.0050+j0.0012 |
| 3 | 0 | 0 | 3–4 | 0.0015+j0.0036 |
| 4 | 0 | 0 | 4–5 | 0.0251+j0.0294 |
| 5 | 0 | 0 | 5–6 | 0.3660+j0.1864 |
| 6 | 2.6 | 2.2 | 6–7 | 0.3811+j0.1941 |
| 7 | 40.4 | 30 | 7–8 | 0.0922+j0.0470 |
| 8 | 75 | 54 | 8–9 | 0.0493+j0.0251 |
| 9 | 30 | 22 | 9–10 | 0.8190+j0.2707 |
| 10 | 28 | 19 | 10–11 | 0.1872+j0.0691 |
| 11 | 145 | 104 | 11–12 | 0.7114+j0.2351 |
| 12 | 145 | 104 | 12–13 | 1.0300+j0.3400 |
| 13 | 8 | 5.5 | 13–14 | 1.0440+j0.3450 |
| 14 | 8 | 5.5 | 14–15 | 1.0580+j0.3496 |
| 15 | 0 | 0 | 15–16 | 0.1966+j0.0650 |
| 16 | 45.5 | 30 | 16–17 | 0.3744+j0.1238 |
| 17 | 60 | 35 | 17–18 | 0.0047+j0.0016 |
| 18 | 60 | 35 | 18–19 | 0.3276+j0.1083 |
| 19 | 0 | 0 | 19–20 | 0.2106+j0.0696 |
| 20 | 1 | 0.6 | 20–21 | 0.3416+j0.1129 |
| 21 | 114 | 81 | 21–22 | 0.0140+j0.0046 |
| 22 | 5.3 | 3.5 | 22–23 | 0.1591+j0.0526 |
| 23 | 0 | 0 | 23–24 | 0.3463+j0.1145 |
| 24 | 28 | 20 | 24–25 | 0.7488+j0.2457 |
| 25 | 0 | 0 | 25–26 | 0.3089+j0.1021 |
| 26 | 14 | 10 | 26–27 | 0.1732+j0.0572 |
| 27 | 14 | 10 | 3–28 | 0.0044+j0.0108 |
| 28 | 26 | 18.6 | 28–29 | 0.0640+j0.1565 |
| 29 | 26 | 18.6 | 29–30 | 0.3978+j0.1315 |
| 30 | 0 | 0 | 30–31 | 0.0702+j0.0232 |
| 31 | 0 | 0 | 31–32 | 0.3510+j0.1160 |
| 32 | 0 | 0 | 32–33 | 0.8390+j0.2816 |
| 33 | 14 | 10 | 33–34 | 1.7080+j0.5646 |
| 34 | 19.5 | 14 | 34–35 | 1.4740+j0.4873 |
| 35 | 6 | 4 | 4–36 | 0.0034+j0.0084 |
| 36 | 0 | 0 | 36–37 | 0.0851+j0.2083 |
| 37 | 79 | 56.4 | 37–38 | 0.2898+j0.7091 |
| 38 | 384.70 | 274.5 | 38–39 | 0.0822+j0.2011 |
| 39 | 384.70 | 274.5 | 8–40 | 0.0928+j0.0473 |
| 40 | 40.5 | 28.3 | 40–41 | 0.3319+j0.1114 |
| 41 | 3.6 | 2.7 | 9–42 | 0.1740+j0.0886 |
| 42 | 4.35 | 3.5 | 42–43 | 0.2030+j0.1034 |
| 43 | 26.4 | 19 | 43–44 | 0.2842+j0.1447 |
| 44 | 24 | 17.2 | 44–45 | 0.2813+j0.1433 |
| 45 | 0 | 0 | 45–46 | 1.5900+j0.5337 |
| 46 | 0 | 0 | 46–47 | 0.7837+j0.2630 |
| 47 | 0 | 0 | 47–48 | 0.3042+j0.1006 |

(*Continued*)

**Table 4.** (Continued)

| Bus number | $P_d$ (kW) | $Q_d$(kVar) | branch from-to | Impedance($\Omega$) |
|---|---|---|---|---|
| 48 | 100 | 72 | 48–49 | 0.3861+j0.1172 |
| 49 | 0 | 0 | 49–50 | 0.5075+j0.2585 |
| 50 | 1244 | 888 | 50–51 | 0.0974+j0.0496 |
| 51 | 32 | 23 | 51–52 | 0.1450+j0.0738 |
| 52 | 0 | 0 | 52–53 | 0.7105+j0.3619 |
| 53 | 227 | 162 | 53–54 | 1.041+j0.5302 |
| 54 | 59 | 42 | 11–55 | 0.2012+j0.0611 |
| 55 | 18 | 13 | 55–56 | 0.0047+j0.0014 |
| 56 | 18 | 13 | 12–57 | 0.7394+j0.2444 |
| 57 | 28 | 20 | 57–58 | 0.0047+j0.0016 |
| 58 | 28 | 20 | 3–59 | 0.0044+j0.0108 |
| 59 | 26 | 18.55 | 59–60 | 0.0640+j0.1565 |
| 60 | 26 | 18.55 | 60–61 | 0.1053+j0.1230 |
| 61 | 0 | 0 | 61–62 | 0.0304+j0.0355 |
| 62 | 24 | 17 | 62–63 | 0.0018+j0.0021 |
| 63 | 24 | 17 | 63–64 | 0.7283+j0.8509 |
| 64 | 1.2 | 1 | 64–65 | 0.3100+j0.3623 |
| 65 | 0 | 0 | 65–66 | 0.0410+j0.0478 |
| 66 | 6 | 4.3 | 66–67 | 0.0092+j0.0116 |
| 67 | 0 | 0 | 67–68 | 0.1089+j0.1373 |
| 68 | 39.22 | 26.3 | 68–69 | 0.0009+j0.0012 |
| 69 | 39.22 | 26.3 | | |

SOC relaxation, the optimal solution S found in $C_{soc}$ is a lower bound solution of the original problem. If the optimal solution is a point in the feasible region $C_{original}$, it is the optimal solution of the original problem. The equal sign in Eq (16) can be guaranteed to be accurate enough to satisfy all constraints of the original problem when the original problem finds the optimal solution.

If the integer variables are taken into account, the original problem is extended to a SOC optimization problem with mixed integer variables. Existing algorithm packages such as CPLEX, Gurobi and MOSEK can find the optimal solution of the original problem by cutting-plane method or branch and bound method.

## 4. Case analysis and discussion

As mentioned above, for the distribution network, we relax the Eq (15) into the Eq (16). The original feasible region is relaxed into a wider feasible region, and then the problem to be solved in the relaxed feasible region has a strong convexity. Thus, we turned the original non-convex optimization problem into a convex problem.

**Table 5.** Optimal solution in modified IEEE 69.

| Solution time (s) | Objective Function: $C_1$*Network loss+ $C_2$*PV abandon (¥) | | |
|---|---|---|---|
| | 0.526*Network loss | 0.474 *PV abandon | Sum |
| 1.87 | 12.22 | 308.37 | 320.59 |

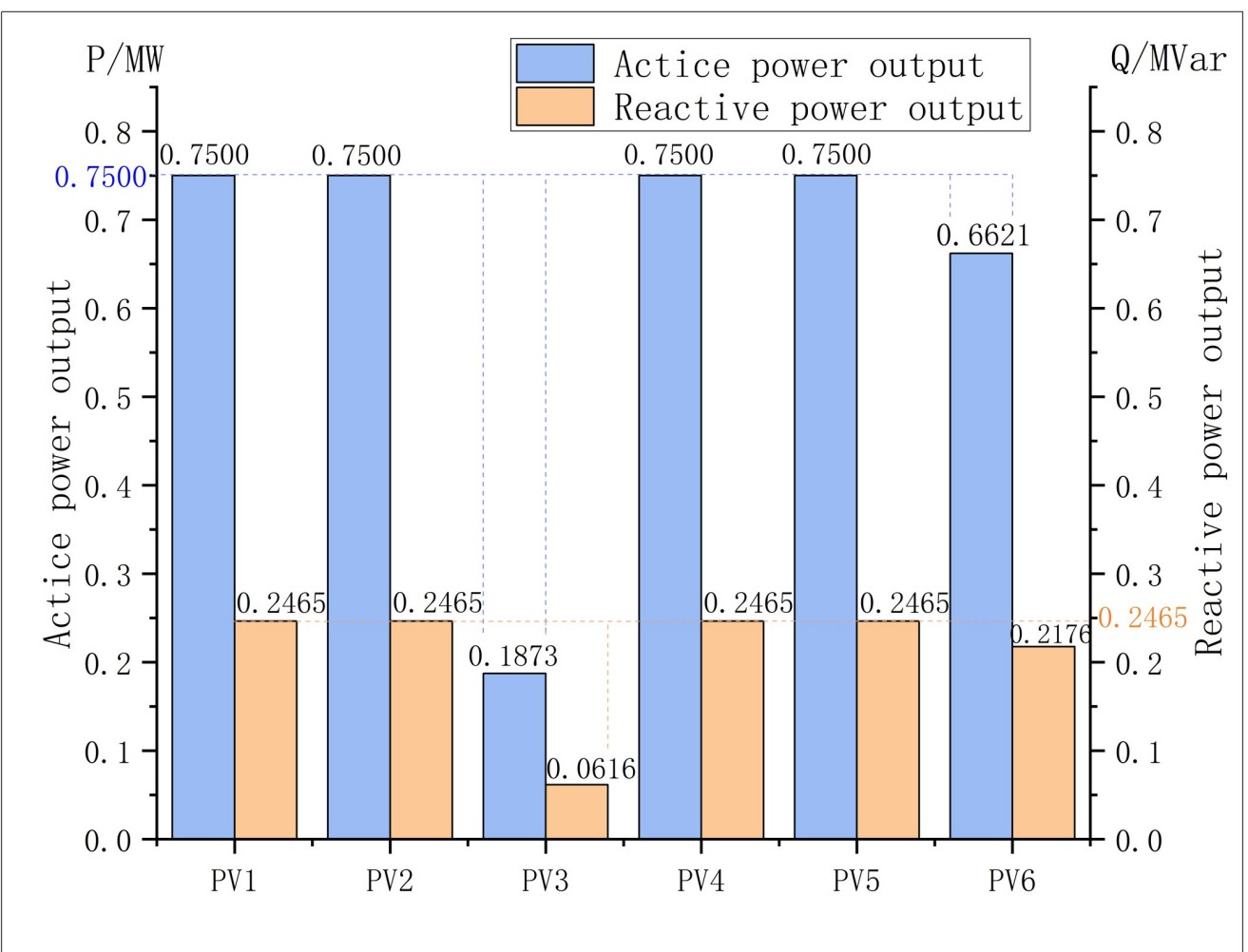

**Fig 11. PVs outputs in modified IEEE 69.**

In order to verify the performance, the above procedure is programmed using MATLAB and algorithm packages such as CPLEX. The hardware environment of the system is AMD Ryzen 7 5800H,CPU@3.2GHz and 8GB memory, with Windows 11 64bit installed as the operating system, and the development environment is MATLAB R2022A.

The feasibility and relaxation accuracy of the proposed method are firstly tested on a smaller scale, that is a modified IEEE33 distribution network, and the optimal scheduling strategy for each adjustable reactive power compensation device is obtained. Then, the model is extended to a modified IEEE 69 distribution network, which contains more branches and nodes, in order to test the accuracy, optimality and efficiency of the proposed method under different test systems.

### 4.1. Modified IEEE 33 distribution network test example

The modified IEEE 33 distribution network is shown in Fig 3.

The system consists of 33 branches and operates radially. The voltage class is 12.66kV, the total active power of the load is 3715kW, and the total reactive power is 2300kVar. Nodes 6,

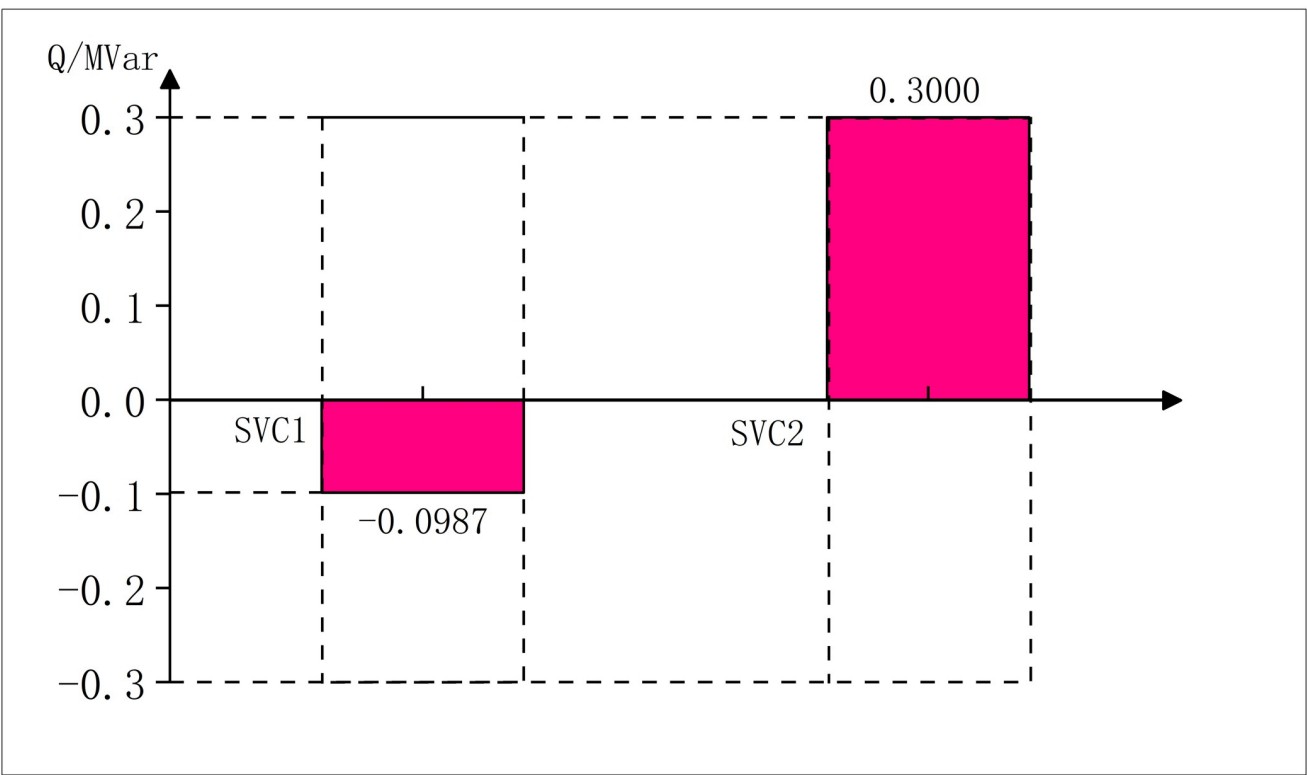

**Fig 12. SVCs outputs in modified IEEE69.**

18, 19 and 31 are respectively connected to PVs. The active power of each PV is set as 1.4MW, and the power factor cos $\varphi$ is 0.95, thus the reactive power is 0.4602 MVar. Given the small geographical distance of each PV, it is considered that the predicted power of the four PVs is equal. Nodes 22 and 25 are respectively connected with CBs, where the maximum step number is 3, and the step size is 100kVar. Node 9 is connected with a continuous adjustable SVC, and the compensation range of SVC is -300kVar to 300kVar.

The load of each bus and the impedance of each branch is shown in Table 2:

Based on the above 33 nodes corresponding to the data and parameters, the solutions and optimal scheduling strategy obtained by calling CPLEX in MATLAB are shown in the following table and figures:

The information about optimal solution is shown in Table 3.

After solving the problem, the minimum value of objective function is 894.99, which represents the minimum economic loss per hour caused by the network loss and PV abandon. It has certain reference value for the comprehensive economic operation of the active distribution network with PVs.

The outputs of PVs, SVC and CBs under the optimal solution are respectively shown in the Figs 4–6:

The proportions of discarded active power and on-gird power of PV are shown in Fig 7.

Voltage distribution is an important index of power quality, the curves of bus voltage amplitude before and after the optimization are shown in Fig 8. There has been a very significant improvement in the bus voltage amplitude after the coordinated active-reactive power optimization.

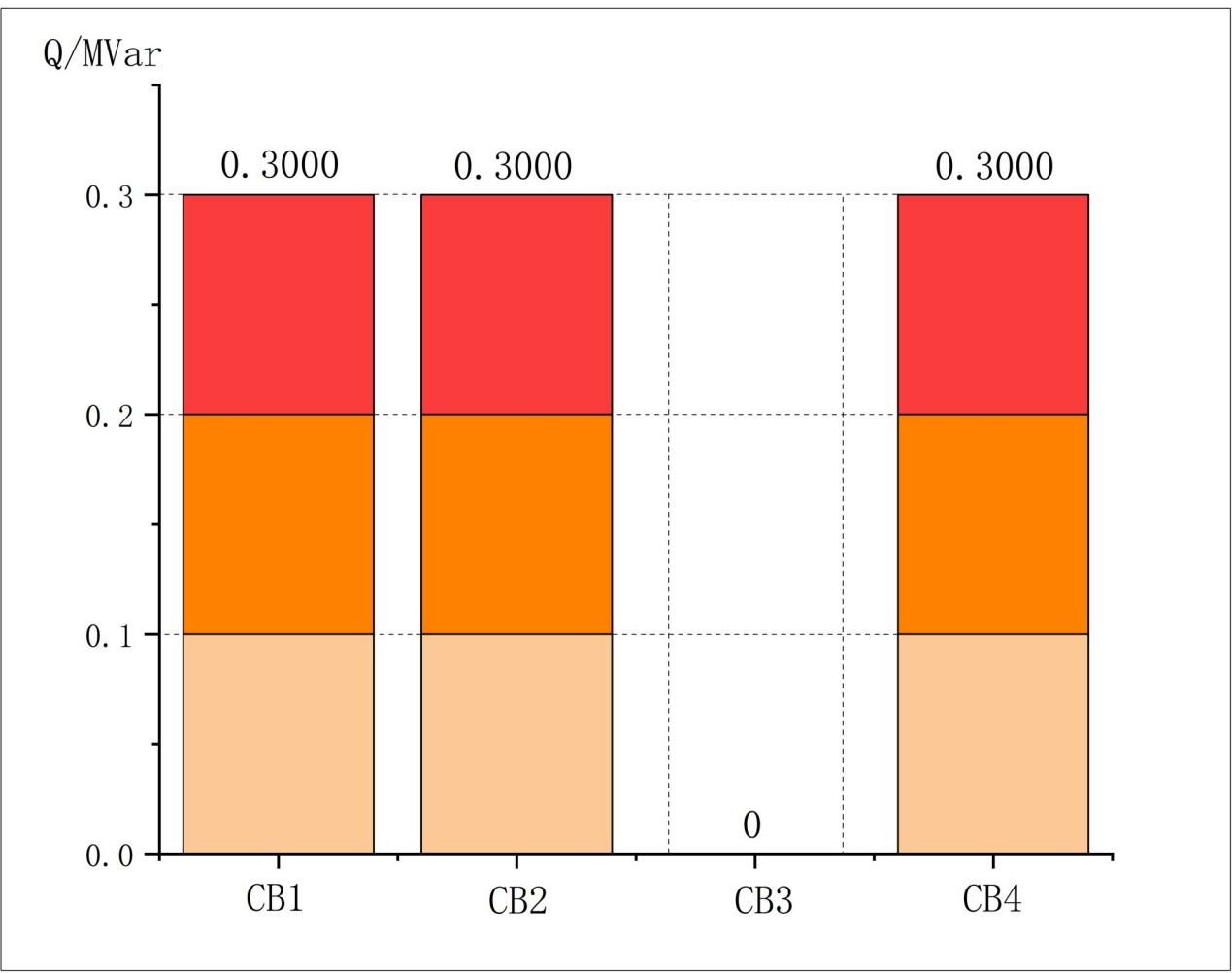

**Fig 13. CBs output in modified IEEE69.**

The example provides a set of optimal solution information for the modified IEEE 33 to operate with the minimum economic loss, and the bus voltage amplitude is also improved.

Finally, we explore the impact of different C1 and C2 in different regions on the optimal solution. By appropriately simulating the results of different regions, we can perform sensitivity analysis on the optimization model. Under the premise of ensuring that the sum of C1 and C2 is 1, C1 starts from 0 and takes a step of 0.1 to explore the changes in the optimal solution under different weighting factors. In addition, we use C1 and C2 as indicators to explore the accuracy of SOCP relaxation, and count the maximum gap value of the model under different weighting factors to verify the accuracy of the model. The results are shown in the Fig 9 below:

By changing the weighting factors of C1 and C2, it can be found that when the weight of C1 increases, the economic loss of the optimal solution usually decreases. This is because the model will consider reducing the transmission line loss more during the optimization process, and the reduction of this part of the loss can bring significant economic benefits. However, this may also lead to an increase in the abandonment loss of photovoltaic modules, because the model may tend to reduce the load of the transmission line and ignore the full utilization of photovoltaic power generation. Therefore, the weight setting of C1 and C2 needs to find a balance between reducing transmission line losses and photovoltaic module abandonment losses.

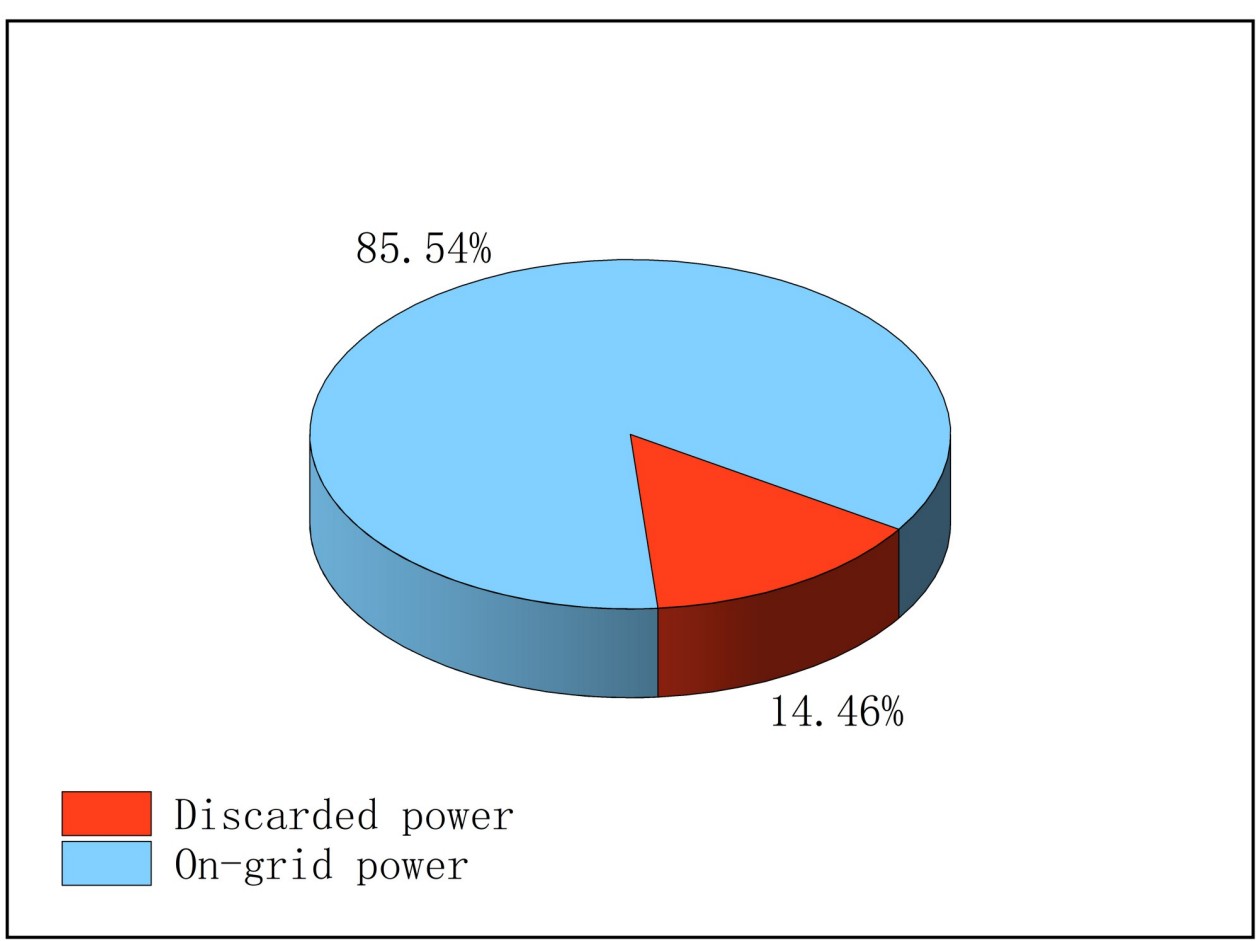

**Fig 14. The proportions of abandoned and on-grid power from all PVs.**

The weighting factor values of C1 and C2 determined in this study are in line with the local situation in Qinghai Province.

We found that with the increase of C1, that is, the increase in the proportion of network loss, the relaxation gap tends to decrease as a whole, and a sharp drop occurs at 0.5. In this study, C1 is set to be 0.526, and the corresponding maximum relaxation gap is $1.23 \times 10^{-6}$. This indicates that network loss is the main influencing factor of the relaxation gap. Under the premise of fixed resistance, small network loss means relatively small current, so the difference in the relaxation gap will also be smaller. In the weighting factors selected in this study, the relaxation gap is very small, which can indirectly prove that our simulation optimization model has high accuracy.

## 4.2. Modified IEEE 69 distribution network test example

The modified IEEE 69 distribution network is shown in Fig 10.

The system consists of 69 branches and operates radially. The voltage class is 12.66kV, the total active power of the load is 3715kW, and the total reactive power is 2300kVar. Nodes 7, 12, 35, 50, 53 and 69 are respectively connected to PVs. The active power of each PV is set as 0.75MW, and the power factor cos $\varphi$ is 0.95, thus the reactive power is 0.2465MVar. Given the

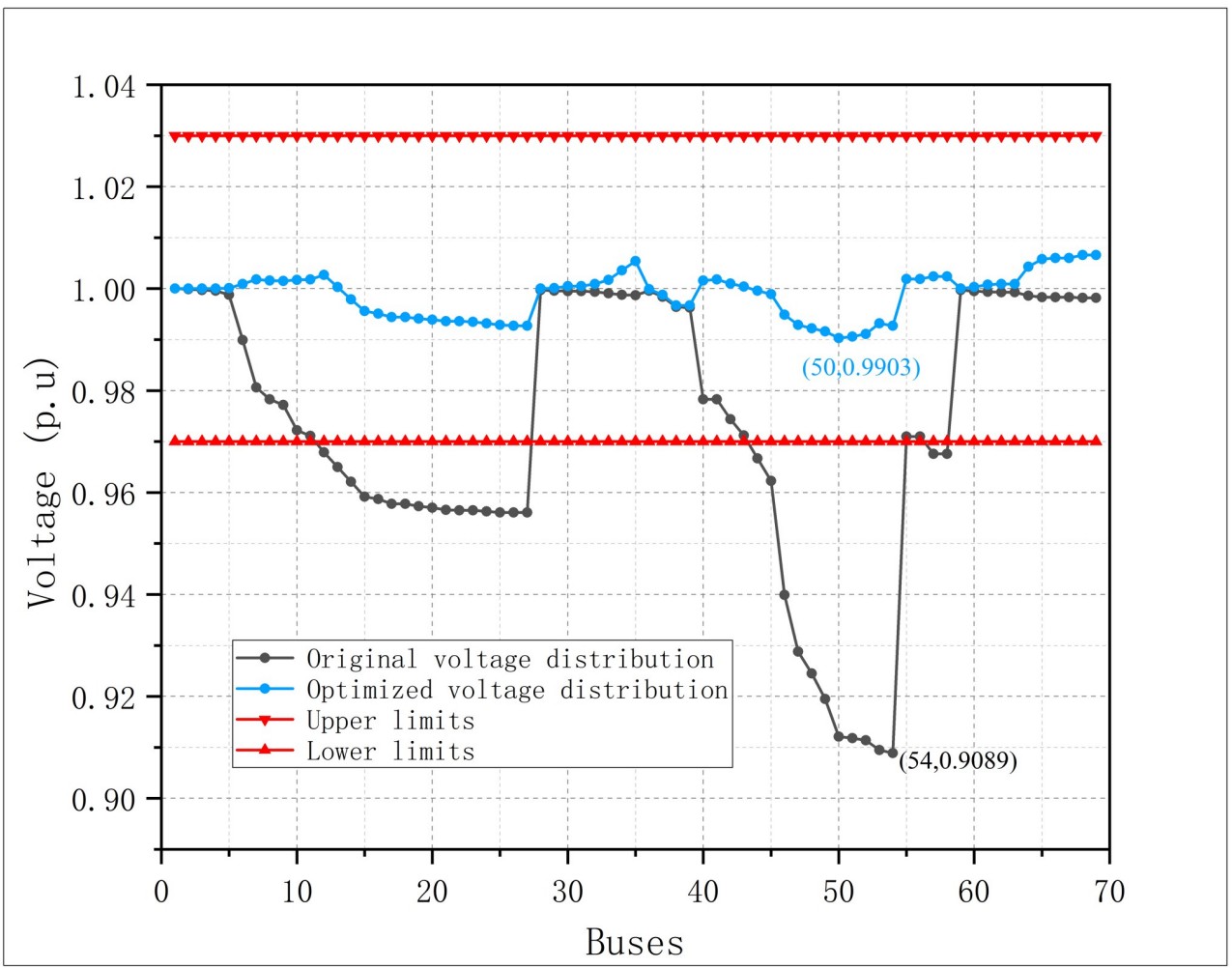

**Fig 15. Bus voltage amplitude before and after optimization in modified IEEE 69.**

small geographical distance of each PV, it is considered that the predicted PV power of the six PVs is equal. Nodes 21, 41, 57 and 56 are respectively connected with four CBs, where the maximum compensation of each CB is 300kVar, and the step size is 100kVar. Nodes 27 and 45 are connected with two SVCs, and the compensation range of each SVC is -300kVar to 300kVar. Based on the above data and configuration parameters, the solutions and optimal scheduling strategy are shown in the following table and figure. The method of solving this case is consistent with that in modified IEEE33.

The load of each bus and the impedance of each branch is shown in Table 4:

The information about optimal solution is shown in Table 5.

After solving the problem, the minimum value of objective function is 320.59, which represents the minimum economic loss per hour caused by the network loss and PV abandon, and the outputs of PVs, SVCs and CBs in the optimal solution are respectively shown in the Figs 11–13:

The proportions of discarded active power and on-gird power of PVs are shown in Fig 14.

The curves of bus voltage amplitude before and after the optimization are shown in Fig 15. It can be seen that the bus voltage amplitude also has a significant improvement after the coordinated active-reactive power optimization.

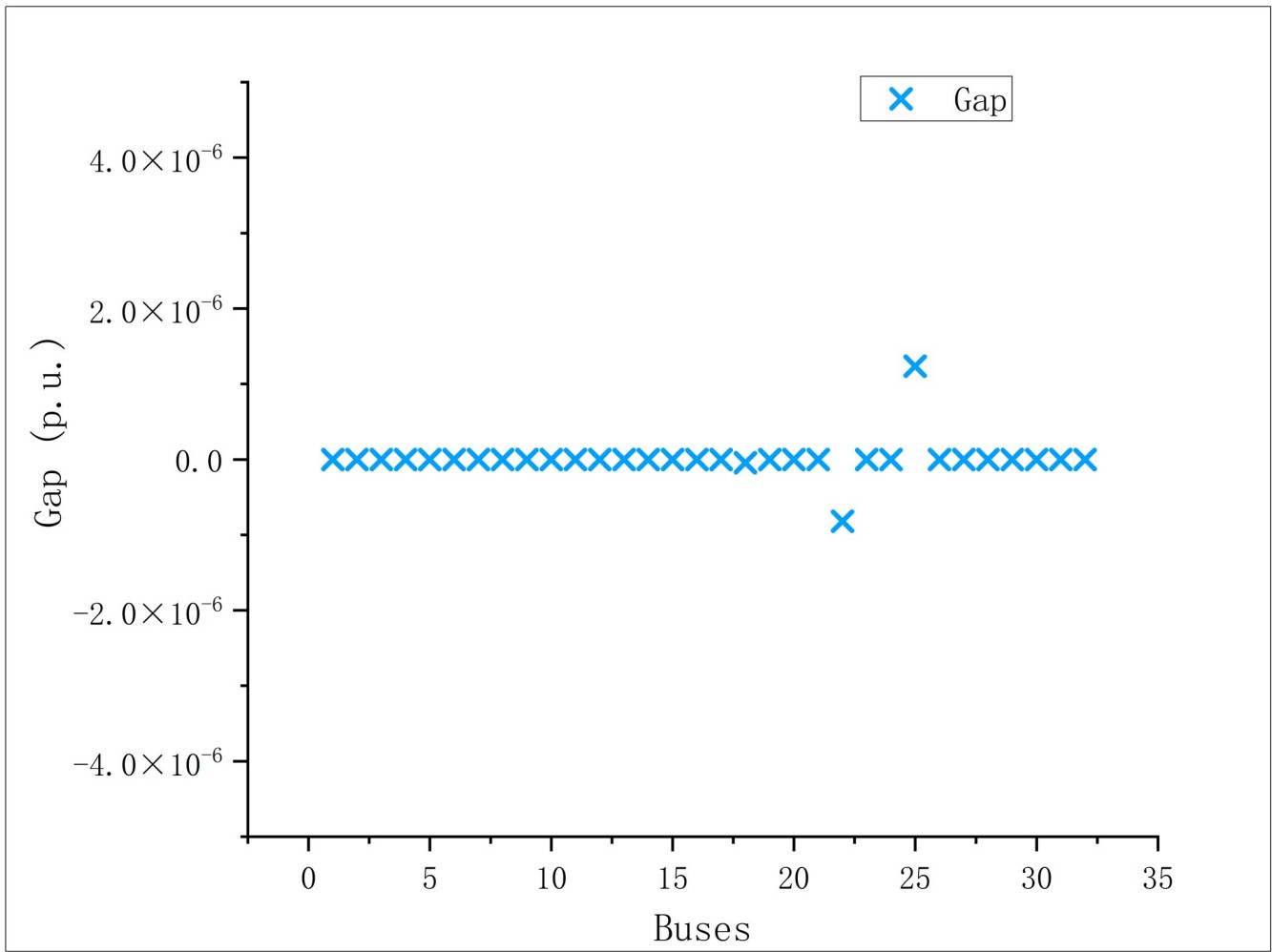

**Fig 16. Scatter diagram of relaxation gap of modified IEEE 33.**

The example provides a set of optimal solution information for the modified IEEE 69 to operate with the minimum economic loss, and the the bus voltage amplitude is also improved.

### 4.3. Accuracy analysis of relaxation

In order to verify the accuracy of Eq (16), the gap distribution caused by SOC relaxation is shown in the Figs 16 and 17:

At the optimal solution after relaxation, the infinite norm of the SOC relaxation gap vector of the branch is defined as follows:

$$devi = \left\| \mathbf{i}_2 - \frac{(\mathbf{P})^2 + (\mathbf{Q})^2}{\mathbf{v}_2} \right\|_\infty \tag{23}$$

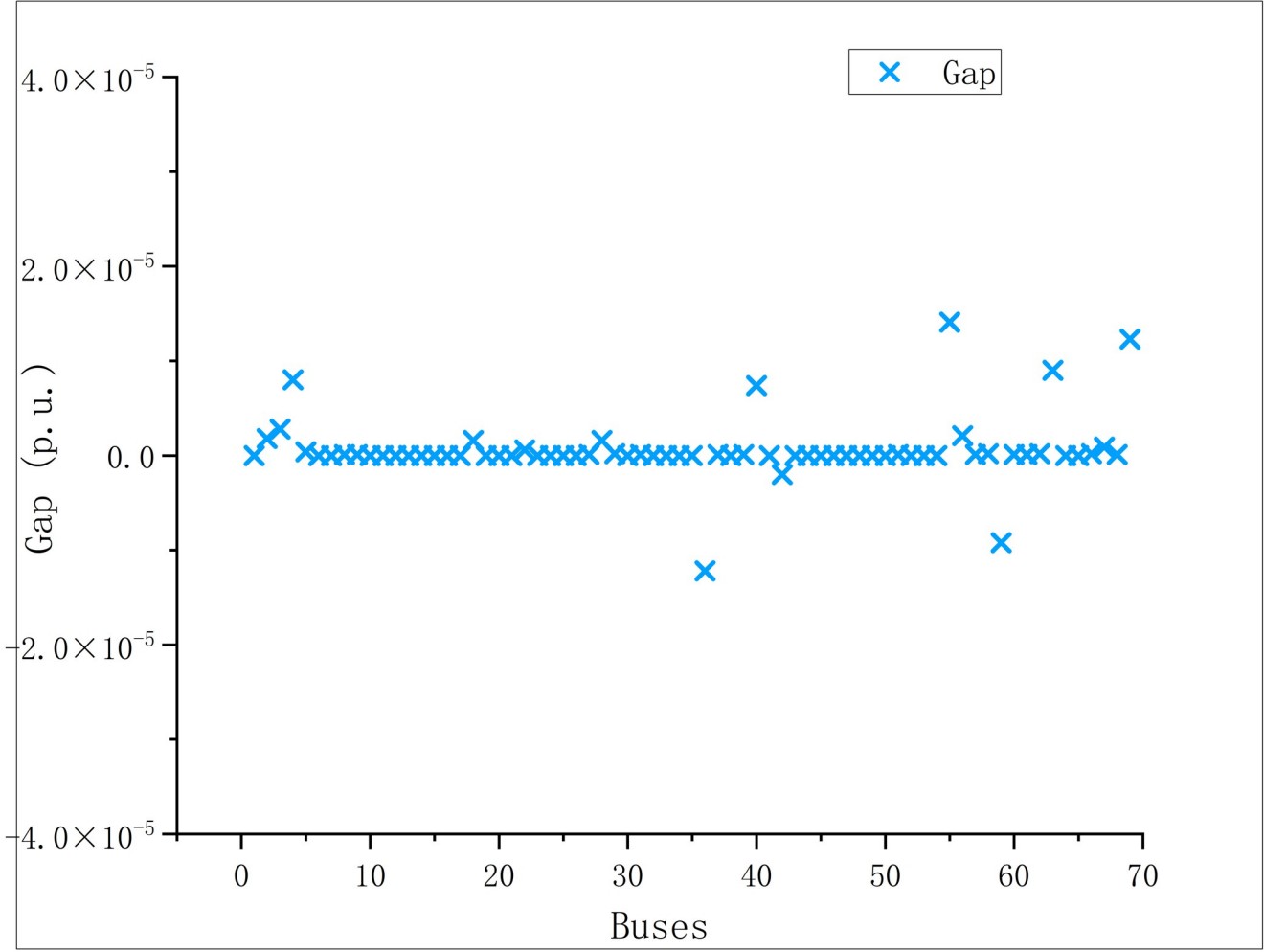

**Fig 17. Scatter diagram of relaxation gap of modified IEEE 69.**

In the modified IEEE 33 optimal solution,

$$devi = 1.236 \times 10^{-6} \tag{24}$$

In the modified IEEE 69 optimal solution,

$$devi = 1.410 \times 10^{-5} \tag{25}$$

It is not difficult to find from the above equation that the SOC relaxation adopted in this paper is accurate enough.

Finally, we randomly generated 10000 sets of data for Monte Carlo algorithm quantization to verify the results. The optimal solution can be obtained only under the configuration conditions of the above solution. It can be seen that the calculation example is valid and reasonable.

## 5. Conclusions

This paper established a DistFlow model suitable for radial distribution networks and introduced SOC relaxation to convert non-convex problems into convex ones, further extending it

to a MISOCP model. The numerical examples verified the method's high relaxation accuracy, short solution time, and strong optimality. Future work plans to study the network topology of radial distribution networks, and then reconstruct and optimize them based on the original ones; then further combine the time scale to perform dynamic planning and reconstruction based on the optimal value of the real-time distribution network. The current study does not account for variables introduced by multiple daily sessions or constraints on the number of switching times of discrete devices. Future research should focus on these aspects to develop a more comprehensive optimized scheduling strategy. Additionally, solving more complex mathematical models will be necessary to address these challenges.

## Supporting information

**S1 File.**
(M)

**S2 File.**
(M)

## Author Contributions

**Conceptualization:** Yongjie Wang.

**Data curation:** Bo Peng.

**Funding acquisition:** Yongjie Wang.

**Methodology:** Yongjie Wang.

**Software:** Bo Peng.

**Validation:** Yongjie Wang.

**Writing – original draft:** Bo Peng.

**Writing – review & editing:** Bo Peng.

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
