## [Decision Letter · Decision Letter 0]

23 May 2024

PONE-D-23-42185Coordinated Active-Reactive Power Optimization Considering Photovoltaic Abandon based on Second Order Cone Programming in Active Distribution NetworksPLOS ONE

Dear Dr. wang,

Thank you for submitting your manuscript to PLOS ONE. After careful consideration, we feel that it has merit but does not fully meet PLOS ONE’s publication criteria as it currently stands. Therefore, we invite you to submit a revised version of the manuscript that addresses the points raised during the review process.

We look forward to receiving your revised manuscript.

Kind regards,

Mohamad Abou Houran

Academic Editor

PLOS ONE

Journal Requirements:

"NO"

Additional Editor Comments:

Thank you for submitting your manuscript to PLOS ONE. The reviewers have completed the review of your manuscript, major revisions are required before considering the publication of your work. If you are able to address the reviewers' comments well, we ask you to respond to each comment by either outlining how the criticism was addressed in the revised manuscript or by providing a rebuttal to the criticism.

Reviewers' comments:

Reviewer's Responses to Questions

**Comments to the Author**

1. Is the manuscript technically sound, and do the data support the conclusions?

Reviewer #1: Yes

Reviewer #2: Partly

2. Has the statistical analysis been performed appropriately and rigorously? 

Reviewer #1: Yes

Reviewer #2: Yes

3. Have the authors made all data underlying the findings in their manuscript fully available?

Reviewer #1: Yes

Reviewer #2: Yes

4. Is the manuscript presented in an intelligible fashion and written in standard English?

Reviewer #1: Yes

Reviewer #2: Yes

5. Review Comments to the Author

Reviewer #1: The paper entitled Coordinated Active-Reactive Power Optimization Considering Photovoltaic Abandon based on Second Order Cone Programming in Active Distribution Networks

• The paper is well organised

• Did authors have considered the uncertainties in load. Justify it.

• Why authors considered 1 and 2, as 0.526 and 0.474. Justify it.

• Why authors have selected specific buses for PV and CBs installation? Justify it

• More recent literature could be included especially from relevant publications

Reviewer #2: I have now completed the review and these are the suggestions.

Major Comments:

The introduction section needs to be more focused and provide a clear motivation for the work presented. The background information on distributed generation and reactive power optimization is extensive, but the specific gap or problem addressed by this study should be clearly highlighted.

Include these works in literature review:

Khan NM, Khan UA, Asif M, Zafar MH. Analysis of deep learning models for estimation of MPP and extraction of maximum power from hybrid PV-TEG: A step towards cleaner energy production. Energy Reports. 2024 Jun 1;11:4759-75.

Al-Tawalbeh N, Zafar MH, Radzi MA, Zainuri MA, Al-Wesabi I. Novel initialization strategy: Optimizing conventional algorithms for global maximum power point tracking. Results in Engineering. 2024 Jun 1;22:102067.

Mansoor M, Abou Houran M, Al-Tawalbeh N, Zafar MH, Akhtar N. Thermoelectric power generation system intelligent Runge Kutta control: A performance analysis using processor in loop testing. Energy Conversion and Management: X. 2024 May 1:100612.

The objective function formulation needs further clarification and justification. The rationale behind the coefficients C1 and C2, and the inclusion of PV abandon and network loss components, should be explained in more detail.

The constraint formulation section could be improved by providing a clearer explanation of the various constraints considered, such as voltage limits, branch current limits, and reactive power compensation device constraints.

The solution methodology section could benefit from a more detailed explanation of the Second-Order Cone Programming (SOCP) technique used. The mathematical derivations and relaxations involved should be explained more clearly, especially for readers unfamiliar with SOCP.

The case study section should provide more information on the modified IEEE 33 and IEEE 69 distribution networks used for testing. Details such as network topology, load profiles, and DG/PV locations should be clearly stated.

The results and discussion section could be improved by providing a more comprehensive analysis of the obtained solutions. The impact of different weighting factors (C1 and C2) on the optimal solution and the trade-off between network loss and PV abandon should be discussed.

The accuracy analysis of the relaxation technique used should be more thorough. Additional metrics or statistical measures could be used to quantify the accuracy and potential errors introduced by the relaxation.

The conclusions section should be more concise and focused on the key findings and contributions of the study. Potential limitations and future work directions should also be discussed.

Minor Comments:

Some of the equations and variable definitions could be improved for better readability and clarity. For example, the use of subscripts and superscripts should be consistent throughout the manuscript.

The abbreviations and acronyms used in the manuscript should be defined at their first occurrence or in a separate list of abbreviations.

The literature review section could be more comprehensive and include more recent references related to the problem addressed in the study.

The quality of some figures, particularly those showing network topologies, could be improved for better clarity and visual representation.

The manuscript could benefit from a thorough language and grammar check to improve the overall readability and clarity of the text.

6. PLOS authors have the option to publish the peer review history of their article (what does this mean?). If published, this will include your full peer review and any attached files.

Reviewer #1: **Yes: **Dr. Pamshetti Vijay Babu

Reviewer #2: No

---

## [Author Response · Author response to Decision Letter 0]

10 Jul 2024

First, we would like to sincerely thank the referee for the insightful comments and invaluable suggestions. Here are the responses to the reviewer’s comments. We have numbered the reviewer’s comments to facilitate referring to them in the manuscript.(The word version of the reviewer's comments is also uploaded as an attachment in the system.)

Responses to Referee 1:

Comment1：Did authors have considered the uncertainties in load. Justify it.

 This study sets the time scale to a certain time section or a small time window, and simulates and optimizes the distribution network at this moment. Therefore, this paper does not discuss the uncertainty of the load, but we plan to conduct research related to dynamic programming and network reconstruction based on this paper, and then we will consider multiple time scales and focus on the uncertainty of the load.

Comment2：Why authors considered 1 and 2, as 0.526 and 0.474. Justify it.

 In order to make the C1 and C2 coefficients easier to understand, we added an explanation of the origin of the network loss and abandoned light coefficients in lines 199-209 of the manuscript. We chose this coefficient, which is fundamentally based on the local electricity price situation and the photovoltaic industry and its subsidies in Qinghai Province, China.

Comment3：Why authors have selected specific buses for PV and CBs installation? Justify it

 In fact, the allocation of specific DG and other devices based on the IEEE 33 and IEEE 69 examples is optional. As long as it is reasonable, different authors can adjust the number, location and even parameters of the devices according to the actual conditions in different regions. We just used such a matching method and conducted our simulation optimization research based on this custom method to reflect the superiority of the mathematical model processing and optimization method application used in our paper.

Comment4：More recent literature could be included especially from relevant publications

 Following your suggestion, I have cited two recent articles on electrical engineering published by PLOS ONE in my manuscript. The text is located at lines 33-37, and the citation is located at lines 541-544..

Responses to Referee 2:

Comment 1：The introduction section needs to be more focused and provide a clear motivation for the work presented. The background information on distributed generation and reactive power optimization is extensive, but the specific gap or problem addressed by this study should be clearly highlighted.

 We have revised the review to clarify the research motivation and current research questions, which are located on pages 104-115.

Comment 2：Include these works in literature review:

（1）Khan NM, Khan UA, Asif M, Zafar MH. Analysis of deep learning models for estimation of MPP and extraction of maximum power from hybrid PV-TEG: A step towards cleaner energy production. Energy Reports. 2024 Jun1;11:4759-75.

（2）Al-Tawalbeh N, Zafar MH, Radzi MA, Zainuri MA, Al-Wesabi I. Novel initialization strategy: Optimizing conventional algorithms for global maximum power point tracking. Results in Engineering. 2024 Jun 1;22:102067.

（3）Mansoor M, Abou Houran M, Al-Tawalbeh N, Zafar MH, Akhtar N. Thermoelectric power generation system intelligent Runge Kutta control: A performance analysis using processor in loop testing. Energy Conversion and Management: X. 2024 May 1:100612.

 Your suggestion is very useful, which fills in the weak points in our review, which is on pages 43-46 of the main text. Reference at：590-602.

In order to better enrich the review, we have added three additional articles, which are located in lines 64-66 of the manuscript. The specific added works are:

（1）M. H. Zafar, U. A. Khan and N. M. Khan, "A sparrow search optimization algorithm based MPPT control of PV system to harvest energy under uniform and non-uniform irradiance," 2021 International Conference on Emerging Power Technologies (ICEPT), Topi, Pakistan, 2021, pp. 1-6

（2）Muhammad Hamza Zafar, Noman Mujeeb Khan, Adeel Feroz Mirza, Majad Mansoor,Bio-inspired optimization algorithms based maximum power point tracking technique for photovoltaic systems under partial shading and complex partial shading conditions,Journal of Cleaner Production,Volume 309,2021, 127279

（3）Muhammad Hamza Zafar, Noman Mujeeb Khan, Adeel Feroz Mirza, Majad Mansoor, Naureen Akhtar, Muhammad Usman Qadir, Nauman Ali Khan, Syed Kumayl Raza Moosavi,A novel meta-heuristic optimization algorithm based MPPT control technique for PV systems under complex partial shading condition,Sustainable Energy Technologies and Assessments,Volume 47,2021,101367.

Comment 3：The objective function formulation needs further clarification and justification. The rationale behind the coefficients C1 and C2, and the inclusion of PV abandon and network loss components, should be explained in more detail.

 In response to your point of view, we have modified the expression of the objective function and explained in more detail the meaning of the two parts of the objective function: abandoned light and network loss, as well as the reason for choosing these two as representatives. The specific modified text is located in lines 162-189 of the manuscript.

Comment 4：The constraint formulation section could be improved by providing a clearer explanation of the various constraints considered, such as voltage limits, branch current limits, and reactive power compensation device constraints.

 A new text explanation section has been added to the manuscript for each constraint, which can make it easier for readers to understand the composition of the constraint, and specific constraints such as voltage and current have been added. The specific modified text is located in lines 224-279 of the manuscript.

Comment 5：The solution methodology section couldbenefit from a more detailed explanation of the Second-Order Cone Programming (SOCP) technique used. The mathematical derivations and relaxations involved should be explained more clearly, especially for readers unfamiliar with SOCP.

 On pages 295-313, before the mathematical explanation of the second-order cone model, a new paragraph is added to introduce the SOCP technology in a more popular and straightforward manner. By explaining its principles and relaxation process in a more superficial way, readers can more easily understand the subsequent SOCP content.

Comment 6：The case study section should provide more information on the modified IEEE 33 and IEEE 69 distribution networks used for testing. Details such as network topology, load profiles, and DG/PV locations should be clearly stated.

 The manuscript originally contained detailed information, but we later deleted it after considering the length of the article. After listening to your suggestions, we restored it to the manuscript, which can be found in: lines 411-413 and 474-476, showing the revised network data of IEEE 33 nodes and 69 nodes respectively. 

Comment 7：The results and discussion section could be improved by providing a more comprehensive analysis of the obtained solutions. The impact of different weighting factors (C1 and C2) on the optimal solution and the trade-off between network loss and PV abandon should be discussed.

 After listening to your suggestions, we chose to discuss the impact of different weighting factors in the IEEE33 example. We used different C1C2 weighting factors to perform simulation optimization, statistics and drawing analysis. Finally, we also summarized some rules, and the network loss coefficient factor is inversely proportional to the optimal solution. Located on pages 435-459.

Comment 8：The accuracy analysis of the relaxation technique used should be more thorough. Additional metrics or statistical measures could be used to quantify the accuracy and potential errors introduced by the relaxation.

In solving your Comment 7, we decided to use the change of weighting factors to statistically analyze the maximum node relaxation gap value of the optimized model with different weighting factors. Through statistical calculation, we found that the results are of reference significance. With the increase of C1 (C2=1-C2), the relaxation gap tends to decrease, and it drops three orders of magnitude from 0.007 at about 0.51 and then stabilizes. The maximum gap of the node at 0.526 set in this paper is 1.23e-6 (p.u.). In comparison, the relaxation gap is small, which can prove that the research optimization simulation has high accuracy. The specific location is the same as comment 7, located on pages 435-459.

Comment 9：The conclusionssection should be more concise and focused on the key findings and contributions of the study. Potential limitations and future work directions should also be discussed.

 According to your suggestion, we have revised the original conclusion section and re-edited it to ensure that the content is not redundant and to focus more on its findings, limitations and future research prospects. Located on pages 511-521.

Comment 10：Minor Comments:

（1）Some of the equations and variable definitions could be improved for better readability and clarity. For example, the use of subscripts and superscripts should be consistent throughout the manuscript.

（2）The abbreviations and acronyms used in the manuscript should be defined at their first occurrence or in a separate list of abbreviations.

（3）The literature review section could be more comprehensive and include more recent references related to the problem addressed in the study.

（4）The quality of some figures, particularly those showing network topologies, could be improved for better clarity and visual representation.

（5）The manuscript could benefit from a thorough language and grammar check to improve the overall readability and clarity of the text.

 Thank you for your additional suggestions, which are also very important to us. We checked and regulated the equations in the manuscript to ensure that they are clear and readable. We ensured the abbreviations of special terms were set, and after confirmation, all abbreviations were fully defined and explained when they first appeared. In addition to adding the corresponding six articles, we also added two recent articles in the electrical field of PLOSONE and updated the references. We improved the clarity of the exported images to ensure that you can observe the result data more clearly. Finally, we checked the grammar of the entire article to eliminate potential errors and prevent the possibility of misunderstanding by future readers.

---

## [Decision Letter · Decision Letter 1]

24 Jul 2024

Coordinated Active-Reactive Power Optimization Considering Photovoltaic Abandon based on Second Order Cone Programming in Active Distribution Networks

PONE-D-23-42185R1

Dear Dr. Wang,

We’re pleased to inform you that your manuscript has been judged scientifically suitable for publication and will be formally accepted for publication once it meets all outstanding technical requirements.

Kind regards,

Mohamad Abou Houran

Academic Editor

PLOS ONE

Additional Editor Comments (optional):

Dear authors,

We’re pleased to inform you that your manuscript is judged scientifically suitable for publication in PLOS ONE.

Reviewers' comments:

Reviewer's Responses to Questions

**Comments to the Author**

1. If the authors have adequately addressed your comments raised in a previous round of review and you feel that this manuscript is now acceptable for publication, you may indicate that here to bypass the “Comments to the Author” section, enter your conflict of interest statement in the “Confidential to Editor” section, and submit your "Accept" recommendation.

Reviewer #1: All comments have been addressed

Reviewer #2: All comments have been addressed

2. Is the manuscript technically sound, and do the data support the conclusions?

Reviewer #1: Yes

Reviewer #2: Yes

3. Has the statistical analysis been performed appropriately and rigorously? 

Reviewer #1: N/A

Reviewer #2: Yes

4. Have the authors made all data underlying the findings in their manuscript fully available?

Reviewer #1: Yes

Reviewer #2: Yes

5. Is the manuscript presented in an intelligible fashion and written in standard English?

Reviewer #1: Yes

Reviewer #2: Yes

6. Review Comments to the Author

Reviewer #1: The manuscript titled "Coordinated Active-Reactive Power Optimization Considering Photovoltaic Abandon based on Second Order Cone Programming in Active Distribution Networks".

The authors have addressed all my queries

Reviewer #2: The authors have addressed all the comments. I think paper is now ready to be accepted. Congratulations

7. PLOS authors have the option to publish the peer review history of their article (what does this mean?). If published, this will include your full peer review and any attached files.

Reviewer #1: **Yes: **VIJAY BABU PAMSHETTI

Reviewer #2: No

---

## [Editor Report · Acceptance letter]

31 Jul 2024

PONE-D-23-42185R1 

PLOS ONE

Dear Dr. wang, 

I'm pleased to inform you that your manuscript has been deemed suitable for publication in PLOS ONE. Congratulations! Your manuscript is now being handed over to our production team.

Kind regards, 

on behalf of

Dr. Mohamad Abou Houran 

Academic Editor

PLOS ONE